# Decomposition Polyhedra of Piecewise Linear Functions

**Marie-Charlotte Brandenburg**
Ruhr-Universität Bochum
`marie-charlotte.brandenburg@rub.de`

**Moritz Grillo**
Technische Universität Berlin
`grillo@math.tu-berlin.de`

**Christoph Hertrich**
University of Technology Nuremberg
`christoph.hertrich@utn.de`

## Abstract

In this paper we contribute to the frequently studied question of how to decompose a continuous piecewise linear (CPWL) function into a difference of two convex CPWL functions. Every CPWL function has infinitely many such decompositions, but for applications in optimization and neural network theory, it is crucial to find decompositions with as few linear pieces as possible. This is a highly challenging problem, as we further demonstrate by disproving a recently proposed approach by Tran & Wang (2024). To make the problem more tractable, we propose to fix an underlying polyhedral complex determining the possible locus of nonlinearity. Under this assumption, we prove that the set of decompositions forms a polyhedron that arises as intersection of two translated cones. We prove that irreducible decompositions correspond to the bounded faces of this polyhedron and minimal solutions must be vertices. We then identify cases with a unique minimal decomposition, and illustrate how our insights have consequences in the theory of submodular functions. Finally, we improve upon previous constructions of neural networks for a given convex CPWL function and apply our framework to obtain results in the nonconvex case.

## 1 Introduction

Continuous piecewise linear (CPWL) functions play a crucial role in optimization and machine learning. While they have traditionally been used to describe problems in geometry, discrete and submodular optimization, or statistical regression, they recently gained significant interest as functions represented by neural networks with rectified linear unit (ReLU) activations (Arora et al., 2018). Extensive research has been put into understanding which neural network architectures are capable of representing which CPWL functions (Chen et al., 2022; Haase et al., 2023; Hertrich et al., 2021). A major source of complexity in all the aforementioned fields is nonconvexity. Indeed, not only are nonconvex optimization problems generally much harder to solve than convex ones, but also for neural networks, nonconvexities are usually responsible for making the obtained representations complicated.

It is a well-known folklore fact that every (potentially nonconvex) CWPL function $f \colon \mathbb{R}^n \to \mathbb{R}$ can be written as the difference $f = g - h$ of two *convex* CPWL functions (Melzer, 1986; Kripfganz & Schulze, 1987). Consequently, a natural idea to circumvent the challenges induced by nonconvexity is to use such a decomposition $f = g - h$ and solve the desired problem separately for $g$ and $h$. This is the underlying idea of many successful optimization routines, known as DC programming (see survey by Le Thi & Pham Dinh (2018)), and also occurs in the analysis of neural networks (Zhang et al., 2018). However, the crucial question arising from this strategy is: how much more complex are $g$ and $h$ compared to $f$? A well-established measure for the complexity of a CPWL function is the number of its linear pieces. Therefore, the main question we study in this article is the following.

**Problem 1.1.** *How to decompose a CPWL function $f$ into a difference $f = g - h$ of two convex CPWL functions with as few pieces as possible?*

There exist many ways in the literature to obtain such a decomposition, as we discuss later, but none of them guarantees minimality or at least a useful bound on the number of pieces of $g$ and $h$ depending on those of $f$. In fact, no finite procedure is known that guarantees to find a minimal decomposition, despite a recent attempt by Tran & Wang (2024).

## 1.1 Our Contributions

In this article, we propose a novel perspective on Problem 1.1 making use of polyhedral geometry and prove a number of structural results. We then apply our approach to existing decompositions in the literature, as well as to the theory of submodular functions and to the construction of neural networks representing a given CPWL function, serving as an additional motivation. Our detailed contributions are outlined as follows.

**Decomposition Polyhedra.** After setting the preliminaries in Section 2, Section 3 presents our new polyhedral approach to Problem 1.1. Instead of aiming for a globally optimal decomposition, we propose to restrict to solutions that are *compatible with a given regular polyhedral complex* $\mathcal{P}$. In short, this means fixing where the functions $g$ and $h$ may have breakpoints, that is, points where they are not locally linear. We prove that the set of solutions to decompose $f$ in a way that is compatible with $\mathcal{P}$ is a polyhedron $\mathcal{D}_{\mathcal{P}}(f)$ that arises as the intersection of two shifted polyhedral cones (Theorem 3.5). We call this the *decomposition polyhedron* of $f$ with respect to $\mathcal{P}$. We prove several structural properties of $\mathcal{D}_{\mathcal{P}}(f)$. Among them, we show that the bounded faces of $\mathcal{D}_{\mathcal{P}}(f)$ are exactly those that cannot easily be simplified by subtracting a convex function (Theorem 3.8), and we show that a minimal solution must be a vertex of $\mathcal{D}_{\mathcal{P}}(f)$ (Theorem 3.13). The latter implies a finite procedure to find a minimal decomposition among those that are compatible with $\mathcal{P}$, by simply enumerating the (potentially many) vertices of $\mathcal{D}_{\mathcal{P}}(f)$. It also implies that, if only a single vertex exists, then there is a unique minimal decomposition. We demonstrate that this is indeed the case for important CPWL functions, e.g., the median function, or those computed by a 1-hidden-layer ReLU network.

**Existing Decompositions.** Afterwards, in Section 4, we put our investigations into a broader context within the existing literature. We compare our minimality conditions with existing methods to construct decompositions. Notably, in this context, we refute a conjecture by Tran & Wang (2024), who provide an optimal construction method in dimension 2 and suggest that it might generalize to higher dimensions. We show that it does not.

**Applications to Submodular Function.** In Section 5, we show that our framework entails the setup of set functions which are decomposed into differences of submodular set functions. Representing a set function as such a difference is a popular approach to solve optimization problems similarly to DC programming, see Narasimhan & Bilmes (2005); Iyer & Bilmes (2012); El Halabi et al. (2023). We apply our results from Section 3 to obtain analogous structural insights about (submodular) set functions (Corollary 5.4).

**Application to Neural Network Constructions.** Finally, in Section 6, we study the problem of constructing neural networks representing a given CPWL function. For convex CPWL functions, we blend two incomparable previous constructions by Hertrich et al. (2021) and Chen et al. (2022) to let the user freely choose a trade-off between depth and width of the constructed networks. We then apply the results of this paper to extend this to the nonconvex case by first decomposing the input function as a difference of two convex ones.

**Limitations.** We emphasize that the focus of our paper is fundamental research by building a theoretical foundation to tackle Problem 1.1 and connecting it with other fields. As such, our paper does not imply any direct improvement for a practical task, but it might prove helpful for that in the future. In particular, it is beyond the scope of our paper to provide any implementation of a (heuristic or exact) method to decompose a CPWL function

into a difference of two convex ones. We consider it an exciting avenue for future research to do so, and to apply it to DC programming, discrete optimization, or neural networks.

On the theoretical side, the approach of fixing an underlying compatible polyhedral complex imposes some restriction on the set of possible solutions and can therefore be seen as a limitation. However, we think that this assumption is well justified by the structural properties this assumption allows us to infer and by the examples we demonstrate to fit into the framework. Even with this assumption, the problem remains very challenging.

## 1.2 Further Related Work

Explicit constructions to decompose CPWL functions as differences of convex CPWL functions can be found in several articles, such as Kripfganz & Schulze (1987); Zalgaller (2000); Wang (2004); Schlüter & Darup (2021). This was initiated in the 1-dimensional case by Bittner (1970), and already laid out for positively homogeneous functions in general dimensions by Melzer (1986). Typically, such decompositions are based on certain representations of CPWL functions, which have been constructed, e.g., in Tarela & Martinez (1999); Ovchinnikov (2002); Wang & Sun (2005); see also Koutschan et al. (2023; 2024) for a fresh perspective. These representations also help to understand the representative capabilities of neural networks, see Arora et al. (2018); Hertrich et al. (2021); Chen et al. (2022).

Recently, a minimal decomposition for the 2-dimensional case was given by Tran & Wang (2024). They use a duality between CPWL functions and polyhedral geometry, based on the "balancing condition" from tropical geometry. This condition has already been studied by McMullen (1996) in terms of weight spaces of polytopes. Generally, methods from tropical geometry have been successfully used to understand the geometry of neural networks, see e.g. Zhang et al. (2018); Hertrich et al. (2021); Montúfar et al. (2022); Haase et al. (2023); Brandenburg et al. (2024).

Submodular functions are sometimes called the discrete analogue of convex functions, and optimizing over them is a widely studied problem, which is also relevant for machine learning. A submodular function can be minimized in polynomial time (Grötschel et al., 1981). In analogy to DC programming, this sparked the idea of minimizing a general set function by representing it as a difference of two submodular ones (Narasimhan & Bilmes, 2005; Iyer & Bilmes, 2012; El Halabi et al., 2023). Related decompositions were recently studied by Bérczi et al. (2024). In polyhdral theory, such a decomposition is equivalent to Minkowski differences of generalized permutahedra (Ardila et al., 2009; Jochemko & Ravichandran, 2022).

Another closely related stream of work is concerned with the (exact and approximate) representative capabilities of neural networks, starting with universal approximation theorems (Cybenko, 1989), and specializing to ReLU networks, their number of pieces, as well as depth-width-tradeoffs (Telgarsky, 2016; Eldan & Shamir, 2016; Arora et al., 2018). In addition, Hertrich & Skutella (2023); Hertrich & Sering (2024) provide neural network constructions for CPWL functions related to combinatorial optimization problems. Geometric insights have also proven to be useful to understand the computational complexity of training neural networks (Froese et al., 2022; Bertschinger et al., 2024; Froese & Hertrich, 2024). Recently, Safran et al. (2024) give an explicit construction of how to efficiently approximate the maximum function with ReLU networks.

## 2 Preliminaries

In this section we introduce the necessary preliminaries on polyhedral geometry and CPWL functions. For $m \in \mathbb{N}$, we write $[m] \coloneqq \{1, 2, \ldots, m\}$.

**Polyhedra and Polyhedral Complexes.** A *polyhedron* $P$ is the intersection of finitely many closed halfspaces and a *polytope* is a bounded polyhedron. A hyperplane *supports* $P$ if it bounds a closed halfspace containing $P$, and any intersection of $P$ with such a supporting hyperplane yields a *face* $F$ of $P$. A face is a *proper face* if $F \subsetneq P$ and inclusion-maximal proper faces are referred to as *facets*. A *polyhedral cone* $C \subseteq \mathbb{R}^n$ is a polyhedron such

that $\lambda u + \mu v \in C$ for every $u, v \in C$ and $\lambda, \mu \in \mathbb{R}_{\geq 0}$. The *dual cone* of $C$ is $C^\vee = \{y \in (\mathbb{R}^n)^* \mid \langle x, y \rangle \geq 0 \text{ for all } x \in C\}$. A cone is *pointed* if it does not contain a line. A cone $C$ is *simplicial*, if there are linearly independent vectors $v_1, \ldots, v_k \in \mathbb{R}^n$ such that $C = \{\sum_{i=1}^k \lambda_i v_i \mid \lambda_i \geq 0\}$. For a cone $C$ and $t \in \mathbb{R}^n$, we call $t + C$ a *shifted cone* or *translated cone*.

A *polyhedral complex* $\mathcal{P}$ is a finite collection of polyhedra such that (i) $\emptyset \in \mathcal{P}$, (ii) if $P \in \mathcal{P}$ then all faces of $P$ are in $\mathcal{P}$, and (iii) if $P, P' \in \mathcal{P}$, then $P \cap P'$ is a face both of $P$ and $P'$. For a polyhedral complex $\mathcal{P}$ in $\mathbb{R}^n$, we denote by $\mathcal{P}^d$ the set of $d$-dimensional polyhedra in $\mathcal{P}$. Given two $n$-dimensional polyhedral complexes $\mathcal{P}, \mathcal{Q}$, the complex $\mathcal{P}$ is a *refinement* of $\mathcal{Q}$ if for every $\tau \in \mathcal{P}^n$ there exists $\sigma \in \mathcal{Q}^n$ such that $\tau \subseteq \sigma$. The complex $\mathcal{Q}$ is a *coarsening* of $\mathcal{P}$ if $\mathcal{P}$ is a refinement of $\mathcal{Q}$.

The *star* of a face $\tau \in \mathcal{P}$ is the set of all faces containing $\tau$, i.e., $\text{star}_{\mathcal{P}}(\tau) = \{\sigma \in \mathcal{P} \mid \tau \subseteq \sigma\}$. We only consider *complete* polyhedral complexes, i.e., complexes covering $\mathbb{R}^n$. A *polyhedral fan* is a polyhedral complex in which every polyhedron is a cone (see e.g., Figure 1b).

An $n$-dimensional polyhedral complex can be equipped with a *weight function* $w : \mathcal{P}^{n-1} \to \mathbb{R}$, as we describe as follows. Given a face $\sigma \in \mathcal{P}$, we denote by $\text{aff}(\sigma) \subseteq \mathbb{R}^n$ the unique smallest affine subspace containing $\sigma$. The *relative interior* of $\sigma$ is the interior of $\sigma$ inside the affine space $\text{aff}(\sigma)$. For any dimension $d \leq n$ and any $\tau \in \mathcal{P}^{d-1}, \sigma \in \mathcal{P}^d$ with $\tau \subseteq \sigma$, let $e_{\sigma/\tau} \in \mathbb{R}^n$ be the normal vector of $\tau$ with respect to $\sigma$, that is, the unique vector with length one that is parallel to $\text{aff}(\sigma)$, orthogonal to $\text{aff}(\tau)$, and points from the relative interior of $\tau$ into the relative interior of $\sigma$. A pair $(\mathcal{P}, w)$ forms a *balanced (weighted) polyhedral complex* if the weight function satisfies the *balancing condition* at every $\tau \in \mathcal{P}^{n-2}$ (see Figure 1a):

$$\sum_{\substack{\sigma \in \mathcal{P}^{n-1}: \\ \sigma \supset \tau}} w(\sigma) \cdot e_{\sigma/\tau} = 0.$$

We will see (Lemma 3.2) that considering only faces of codimension 2 indeed makes sense, see also the *structure theorem of tropical geometry* (Maclagan & Sturmfels, 2015).

**Continuous Piecewise Linear Functions.** A continuous function $f : \mathbb{R}^n \to \mathbb{R}$ is called *continuous and piecewise linear* (CPWL), if there exists a polyhedral complex $\mathcal{P}$ such that the restriction of $f$ to each full-dimensional polyhedron $P \in \mathcal{P}^n$ is an affine function. If this condition is satisfied, we say that $f$ and $\mathcal{P}$ are *compatible* with each other.

In line with Chen et al. (2022), we define the *number of pieces $q$* of $f$ to be the smallest possible number $|\mathcal{P}^n|$ of full-dimensional regions of a compatible polyhedral complex $\mathcal{P}$. Note that this requires pieces to be *convex sets*, as they are polyhedra. The function $f$ might realize the same affine function on distinct pieces $P, Q \in \mathcal{P}^n$. To account for that, we define the *number of affine components $k$* to be the the number of different affine functions realized on all the pieces in $\mathcal{P}^n$. Note that this quantity is independent of the choice of the particular compatible complex $\mathcal{P}$.

It holds that $k \leq q \leq k!$ and each of these inequalities can be strict, compare the discussion by Chen et al. (2022). We can assume that $k > n$ (and thus $q > n$) because otherwise $f$ can be written as a composition of an affine projection to a lower dimension $n'$ followed by a CPWL function defined on $\mathbb{R}^{n'}$.

If $f$ is a *convex* CPWL function, then it can be uniquely written as the maximum of finitely many affine functions $f(x) = \max_{i \in [k]} g_i(x)$ such that $k = q$. It follows that there is a *unique coarsest* compatible polyhedral complex $\mathcal{P}_f$, namely the one with $\mathcal{P}_f^n = \{\{x \mid g_i(x) = \max_j g_j(x)\} \mid i \in [k]\}$. In particular, we have $k = q$ if $f$ is convex. We call a polyhedral complex $\mathcal{P}$ *regular*, if there exists a convex CPWL function $f$ such that $\mathcal{P} = \mathcal{P}_f$.

## 3   Decomposition Polyhedra

In this section, we introduce and generally study the main concept of the paper, *decomposition polyhedra*. These polyhedra describe the set of possible decompositions of a CPWL

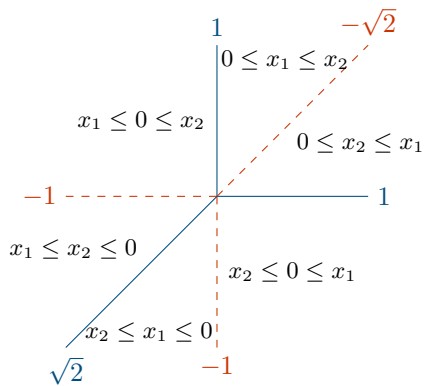
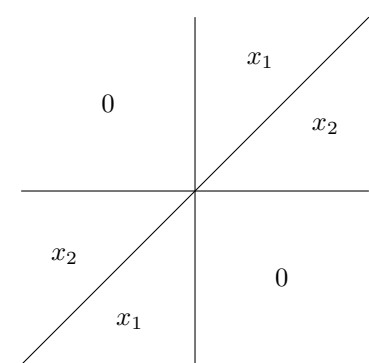

(a) Parameterization of the median function via the weights on the 1-dimensional facets.

(b) Parameterization of the median function via its linear maps on the maximal polyhedra.

Figure 1: Two different parameterizations of the function that computes the median of $\{0, x_1, x_2\}$. This function has $q = 6$ pieces and $k = 3$ affine components, see Example A.1 for more details. In Figure 1a, the convex breakpoints are colored in blue, and concave breakpoints are dashed and colored in red. The absolute value of the weights are given by the euclidean distance of the gradient of the affine components separated by these breakpoints.

function $f$ into a difference $f = g - h$ that are compatible with a given polyhedral complex. We start by establishing a handful of general results concerning the space of CPWL functions compatible with a given polyhedral complex. Proofs that are omitted from the main text as well as some auxiliary statements can be found in Appendix E.

**Lemma 3.1.** *Let $\mathcal{P}$ be a polyhredral complex. The set of CPWL functions compatible with $\mathcal{P}$ forms a linear subspace $\mathcal{V}_\mathcal{P}$ of the space of continuous functions.*

Let $\mathrm{Aff}(\mathbb{R}^n)$ be the space of affine functions from $\mathbb{R}^n$ to $\mathbb{R}$. For many of our arguments, adding or subtracting an affine function $a \in \mathrm{Aff}(\mathbb{R}^n)$ does not change anything. In particular, a function $f$ is convex if and only if $f + a$ is convex. Therefore it makes sense to define the quotient space $\overline{\mathcal{V}}_\mathcal{P} := \mathcal{V}_\mathcal{P} / \mathrm{Aff}(\mathbb{R}^n)$, where we identify functions in $\mathcal{V}_\mathcal{P}$ that only differ by adding an affine function. The following lemma shows that we can parameterize a function $f \in \overline{\mathcal{V}}_\mathcal{P}$ by keeping track of "how convex or concave" the function is at the common face $\sigma \in \mathcal{P}^{n-1}$ of two neighboring pieces. For the case that $w$ is nonnegative and rational, the lemma follows from the *structure theorem of tropical geometry* (Maclagan & Sturmfels, 2015). In Appendix E.2, we present a generalization of the proof adapted to our setting.

**Lemma 3.2.** *The vector space $\mathcal{W}_\mathcal{P} := \{w \colon \mathcal{P}^{n-1} \to \mathbb{R} \mid (\mathcal{P}, w) \text{ is balanced}\}$ is isomorphic to $\overline{\mathcal{V}}_\mathcal{P}$.*

For a function $f \in \mathcal{V}_\mathcal{P}$, let $w_f \in \mathcal{W}_\mathcal{P}$ be the corresponding weight function according to Lemma 3.2. Figure 1 illustrates the different parameterizations of the median function according to Lemma 3.2. Moreover, from Lemma 3.2 we can deduce that $\mathcal{V}_\mathcal{P}$ is finite-dimensional (Corollary E.1) and the following proposition.

**Proposition 3.3.** *A function $f \in \mathcal{V}_\mathcal{P}$ is convex if and only if $w_f$ is nonnegative. Moreover, $f$ is convex with $\mathcal{P} = \mathcal{P}_f$ if and only if $w_f$ is strictly positive.*

The set $\overline{\mathcal{V}}_\mathcal{P}^+$ of *convex* functions in $\overline{\mathcal{V}}_\mathcal{P}$ forms a polyhedral cone (Lemma E.2). In the following, we now fix a function $f \in \mathcal{V}_\mathcal{P}$ and consider the space of decompositions $f = g - h$ into differences of convex functions which are also compatible with $\mathcal{P}$. In Lemma E.3, we show that for a regular complex $\mathcal{P}$ such a decomposition does indeed always exist. In particular, $\overline{\mathcal{V}}_\mathcal{P} = \mathrm{span}(\overline{\mathcal{V}}_\mathcal{P}^+)$, which implies that $\dim(\overline{\mathcal{V}}_\mathcal{P}) = \dim(\overline{\mathcal{V}}_\mathcal{P}^+)$.

**Definition 3.4.** *For a CPWL function $f$ and a polyhedral complex $\mathcal{P}$, the decomposition polyhedron of $f$ with respect to $\mathcal{P}$ is $\mathcal{D}_\mathcal{P}(f) := \{(g, h) \mid g, h \in \overline{\mathcal{V}}_\mathcal{P}^+, f = g - h\}$.*

The projection $\pi((g,h)) = g$ induces an isomorphism between $\mathcal{D}_\mathcal{P}(f)$ and $\pi(\mathcal{D}_\mathcal{P}(f))$ since $\mathcal{D}_\mathcal{P}(f) = \{(g, g-f) \mid g \in \pi(\mathcal{D}_\mathcal{P}(f))\}$. We now show that this is indeed a polyhedron, which arises as the intersection of two shifted copies of a cone.

**Theorem 3.5.** *The set $\mathcal{D}_\mathcal{P}(f)$ is a polyhedron that arises as the intersection of convex functions with an affine hyperplane $H_f = \{(g,h) \mid f = g - h\}$, namely $\mathcal{D}_\mathcal{P}(f) = (\overline{\mathcal{V}}_\mathcal{P}^+ \times \overline{\mathcal{V}}_\mathcal{P}^+) \cap H_f$. Under the bijection $\pi$, the decomposition polyhedron is the intersection of two shifted copies of the polyhedral cone $\overline{\mathcal{V}}_\mathcal{P}^+$. More specifically, $\pi(\mathcal{D}_\mathcal{P}(f)) = \overline{\mathcal{V}}_\mathcal{P}^+ \cap (\overline{\mathcal{V}}_\mathcal{P}^+ + f)$.*

**Remark 3.6.** Under the isomorphism of Lemma 3.2, we identify $\mathcal{D}_\mathcal{P}(f)$ with the polyhedron $\{(w_g, w_h) \in \mathcal{W}_\mathcal{P}^+ \times \mathcal{W}_\mathcal{P}^+ \mid w_g - w_h = w_f\}$ in $\mathcal{W}_\mathcal{P} \times \mathcal{W}_\mathcal{P}$ and $\pi(\mathcal{D}_\mathcal{P}(f))$ with the polyhedron $\{w_g \in \mathcal{W}_\mathcal{P}^+ \mid w_g \geq w_f\}$, where $\mathcal{W}_\mathcal{P}^+ = \{w \in \mathcal{W}_\mathcal{P} \mid w \geq 0\}$.

For the remainder of this section, we analyze the faces of the polyhedron $\overline{\mathcal{V}}_\mathcal{P}^+$ in terms of the properties of the corresponding decompositions.

**Definition 3.7.** A decomposition $(g,h) \in \mathcal{D}_\mathcal{P}(f)$ is called *reduced*, if there is no convex function $\phi \in \overline{\mathcal{V}}_\mathcal{P}^+ \setminus \{0\}$ such that $g - \phi$ and $h - \phi$ are both convex.

If a decomposition is not reduced, then we can obtain a "better" decomposition by simultaneously simplifying both $g$ and $h$ through subtracting a convex function $\phi$. Hence, it makes sense to put a special emphasis on reduced decompositions. Conveniently, the following theorem links this notion to the geometry of $\mathcal{D}_\mathcal{P}(f)$.

**Theorem 3.8.** *A decomposition $(g,h) \in \mathcal{D}_\mathcal{P}(f)$ is reduced if and only if $(g,h)$ is contained in a bounded face of $\mathcal{D}_\mathcal{P}(f)$.*

**Definition 3.9.** We call a convex function $g \in \overline{\mathcal{V}}_\mathcal{P}^+$ a *coarsening* of another convex function $g' \in \overline{\mathcal{V}}_\mathcal{P}^+$ if the unique coarsest polyhedral complex $\mathcal{P}_g$ of $g$ is a coarsening of the unique coarsest polyhedral complex $\mathcal{P}_{g'}$. For a pair of convex CPWL functions $(g,h)$, we call $(g',h')$ a coarsening of $(g,h)$ if $g - h = g' - h'$ and $g$ and $h$ are coarsenings of $g'$ and $h'$ respectively. The coarsening is called non-trivial if $(\mathcal{P}_g, \mathcal{P}_h) \neq (\mathcal{P}_{g'}, \mathcal{P}_{h'})$. For a function $f \in \mathcal{V}_\mathcal{P}$, let $\operatorname{supp}_\mathcal{P}(f) = \{\sigma \in \mathcal{P}^{n-1} \mid w_g(\sigma) \neq 0\}$.

**Lemma 3.10.** *A convex function $g' \in \overline{\mathcal{V}}_\mathcal{P}^+$ is a coarsening of $g \in \overline{\mathcal{V}}_\mathcal{P}^+$ if and only if $\operatorname{supp}_\mathcal{P}(g') \subseteq \operatorname{supp}_\mathcal{P}(g)$. The coarsening is non-trivial if and only if $\operatorname{supp}_\mathcal{P}(g') \subset \operatorname{supp}_\mathcal{P}(g)$.*

**Theorem 3.11.** *Let $(g,h) \in \mathcal{D}_\mathcal{P}(f)$, then the following three statements are equivalent:*

*1. There is no non-trivial coarsening of $(g,h)$.*

*2. $(g,h)$ is a vertex of $\mathcal{D}_\mathcal{P}(f)$.*

*3. $(g,h)$ is a vertex of $\mathcal{D}_\mathcal{Q}(f)$ for all polyhedral complexes $\mathcal{Q}$ compatible with $g$ and $h$.*

**Definition 3.12.** A decomposition $(g,h) \in \mathcal{D}_\mathcal{P}(f)$ is called *minimal*, if it is not *dominated* by any other decomposition, that is, if there is no other decomposition $(g',h') \in \mathcal{D}_\mathcal{P}(f)$ where $g'$ has at most as many pieces as $g$, $h'$ has at most as many pieces as $h$, and one of the two has stricly fewer pieces. See Figure 2 in Appendix A.2 for a visualization.

Phrasing it in terms of multi-objective optimization, we require that the number of pieces of $f$ and $g$ in a minimal decomposition are *Pareto-optimal*. The number of pieces relates to the notion of *monomial complexity* studied in Tran & Wang (2024), and a minimal decomposition translates to a decomposition which is minimal with respect to monomial complexity. We now give a geometric interpretation of this property in terms of $\mathcal{D}_\mathcal{P}(f)$.

**Theorem 3.13.** *A minimal decomposition $(g,h) \in \mathcal{D}_\mathcal{P}(f)$ is always a vertex of $\mathcal{D}_\mathcal{P}(f)$.*

This theorem implies a simple finite procedure to find a minimal decomposition: enumerate all the vertices of $\mathcal{D}_\mathcal{P}(f)$ and choose one satisfying Definition 3.12. It also suggests the following important special case.

**Proposition 3.14.** *If $\mathcal{D}_\mathcal{P}(f)$, or equivalently $\pi(\mathcal{D}_\mathcal{P}(f))$, has a unique vertex, then this vertex corresponds to the unique minimal decomposition within $\mathcal{D}_\mathcal{P}(f)$.*

We now demonstrate that this case is not only convenient, but also it indeed arises for important classes of functions. To this end, recall that $\pi(\mathcal{D}_\mathcal{P}(f)) = \overline{\mathcal{V}}_\mathcal{P}^+ \cap (\overline{\mathcal{V}}_\mathcal{P}^+ + f)$, where

$\overline{\mathcal{V}}_{\mathcal{P}}^{+}$ is a convex, pointed polyhedral cone. In Lemma E.6, we give some sufficient conditions for such intersections of shifted cones to yield a polyhedron with a unique vertex. Moreover, the support of a decomposition can serve as certificate to verify if a decomposition is a unique vertex, and hence minimal. For $f \in \overline{\mathcal{V}}_{\mathcal{P}}^{+}$, let $\operatorname{supp}_{\mathcal{P}}^{+}(f) := \{\sigma \in \mathcal{P} \mid w_f(\sigma) > 0\}$ and $\operatorname{supp}_{\mathcal{P}}^{-}(f) := \{\sigma \in \mathcal{P} \mid w_f(\sigma) < 0\}$.

**Proposition 3.15.** *If for $f \in \overline{\mathcal{V}}_{\mathcal{P}}$, there are $g, h \in \overline{\mathcal{V}}_{\mathcal{P}}^{+}$ such that $f = g - h$ and $\operatorname{supp}^{+}(f) = \operatorname{supp}_{\mathcal{P}}(g)$ as well as $\operatorname{supp}^{-}(f) = \operatorname{supp}_{\mathcal{P}}(h)$, then $(g, h)$ is the unique vertex of $\mathcal{D}_{\mathcal{Q}}(f)$ for every regular complete complex $\mathcal{Q}$ compatible with $f$. In this case, $g$ and $h$ have at most as many pieces as $f$.*

While Proposition 3.15 sounds technical, it is powerful as it allows us to prove that important functions satisfy the condition of Proposition 3.14.

**Definition 3.16.** A *hyperplane function* with $k$ hyperplanes is a function $f \colon \mathbb{R}^n \to \mathbb{R}$ given by $f(x) = \sum_{i \in [k]} \lambda_i \cdot \max\{\langle x, a_i \rangle + b_i, \langle x, c_i \rangle + d_i\}$ for any $a_i, c_i \in \mathbb{R}^n, b_i, d_i, \lambda_i \in \mathbb{R}, i \in [k]$.

Hyperplane functions are precisely the functions that are computable by a ReLU neural network with one hidden layer and appear in this context as 2-term max functions (Hertrich et al. (2021)). They also coincide with functions computable with the hinging hyperplane model (Breiman (1993); Wang & Sun (2005)). Moreover, in Example D.14 we will see that hyperplane functions include continuous extensions of cut functions.

**Definition 3.17.** The *$k$-th order statistic* is the function $f \colon \mathbb{R}^n \to \mathbb{R}$ that returns the $k$-th largest entry of an input vector $x \in \mathbb{R}^n$. For $k = \lfloor \frac{n}{2} \rfloor$, this coincides with the median.

In Appendix A.3, we show that the conditions of Propositions 3.14 and 3.15 are indeed fulfilled for both, hyperplane functions and $k$-th order statistics. This shows that they admit decompositions with at most as many pieces as the function itself.

Theorem 3.8 characterizes the reduced decompositions as bounded faces and Proposition 3.15 provides a condition that can identify a given decomposition as the minimal one. The natural follow-up question is how to find these decompositions. In the following, we show that this can be done via linear programming over the decomposition polyhedron.

**Theorem 3.18.** *A decomposition in $\pi(\mathcal{D}_{\mathcal{P}}(f)) = \overline{\mathcal{V}}_{\mathcal{P}}^{+} \cap (f + \overline{\mathcal{V}}_{\mathcal{P}}^{+})$ is reduced if and only if it is the optimal solution of a linear program with feasible solutions $\pi(\mathcal{D}_{\mathcal{P}}(f))$ and objective linear functional contained in the interior of the dual cone of $\overline{\mathcal{V}}_{\mathcal{P}}^{+}$. In particular, if $\pi(\mathcal{D}_{\mathcal{P}}(f))$ has a single vertex, then the unique optimal solution is the unique reduced and minimal decomposition. Under the isomorphism to $\mathcal{W}_{\mathcal{P}}$, the objective function $u$ can be chosen as $u(\sigma) = 1$ for all $\sigma$.*

## 4    Analysis of existing decompositions

Constructions of decompositions of CPWL functions as difference of two convex functions have appeared in many contexts. In this section, we relate some of these existing constructions to our framework. Moreover, we provide a counterexample to a construction which was proposed by Tran & Wang (2024) to obtain a minimal decomposition in general dimensions.

**Hyperplane extension and local maxima decomposition.**    The literature contains a variety of different constructions to decompose a CPWL function. It is worth noting, however, that these constructions usually follow one of two main themes. The first theme is to construct $(g, h)$ in a way such that they are compatible with the complex $\mathcal{P}$ that arises by extending the codimension-1 faces of $\mathcal{P}_f$ to hyperplanes, see e.g. Zalgaller (2000) and Schlüter & Darup (2021). The second theme is to exploit the properties of the *lattice representation* of a CPWL function (Wang, 2004).

Both of these themes were already illustrated by Kripfganz & Schulze (1987), and we describe their constructions in Appendix B as Construction B.1 ("hyperplane extensions") and Construction B.2 ("local maxima decomposition"). In Appendix B.1, we show that for the functions that compute the $k$-th order statistic, both constructions do not yield the unique minimal decompositions, which exist by Proposition 3.14. This implies the following result.

**Proposition 4.1.** *There is a CPWL function $f$ such that constructions B.1 and B.2 do not provide a vertex of $\mathcal{D}_{\mathcal{P}}(f)$ for any regular polyhedral complex $\mathcal{P}$ compatible with $f$.*

Moreover, Proposition B.3 shows that the suboptimality of the existing decompositions in the literature is not caused by the concrete construction method, but rather by the polyhedral complexes underlying these methods.

**Minimal decompositions.** A construction for a unique minimal decomposition for certain CPWL functions $f$ in dimension 2 was presented by Tran & Wang (2024), by introducing a single new 1-dimensional face to $\mathcal{P}_f$ and an adapted weight function to satisfy the balancing condition.

Their approach builds on a duality theory between *positively homogeneous* convex CPWL functions and *Newton polytopes*. A CPWL function $f$ is positively homogeneous if $f(0) = 0$ and $\mathcal{P}_f$ is a polyhedral fan. Such functions are the *support functions* of their Newton polytopes, as we describe in more detail in Appendix C.1; see also Joswig (2021, Section 1.2) and Maclagan & Sturmfels (2015, Chapter 2).

Based on their 2-dimensional construction, Tran & Wang (2024) propose a procedure to reduce the $n$-dimensional case to 2-dimensions via projections, which we describe in Construction C.2 in Appendix C.2. The final step in this procedure is to construct a global function in $n$ dimensions by "gluing together" the projections. In Example C.3 we illustrate that this final step is not always well-defined. Our conclusion is that the construction by Tran & Wang (2024) does not extend beyond the 2-dimensional case, leaving it an open problem to find any finite algorithm that guarantees to return a minimal decomposition without fixing an underlying polyhedral complex. Note that for any fixed underlying polyhedral complex, Theorem 3.13 implies a finite algorithm by enumerating the vertices of the decomposition polyhedron, as discussed earlier.

## 5 Submodular Functions

We demonstrate that a special case of our framework is to decompose a general set function into a difference of submodular set funtions and translate our results to this setting. Such decompositions are a popular approach to solve optimization problems as disussed in the introduction. Here, we sketch the idea, all details can be found in Appendix D. Let the polyhedral complex $\mathcal{P}$ be induced by the *braid arrangement*, that is, the hyperplane arrangement consisting of the $\binom{n}{2}$ hyperplanes $x_i = x_j$, with $1 \leq i < j \leq n$, and let $\mathcal{F}_n$ be the vector space of set functions from $2^{[n]}$ to $\mathbb{R}$.

**Proposition 5.1.** *The mapping $\Phi$ that maps $f \in \mathcal{V}_{\mathcal{P}}$ to the set function $F(S) = f(\mathbb{1}_S)$, where $\mathbb{1}_S = \sum_{i \in S} e_i$, is a vector space isomorphism.*

Conversely, starting with a set function $F$, then $f = \Phi^{-1}(F)$ is by definition a continuous extension of $F$, which is known as the *Lovász extension* (Lovász, 1983). The Lovász extension is an important concept in the theory and practice of submodular function optimization as it provides a link between *discrete* submodular functions and *continuous* convex functions.

**Definition 5.2.** A set function $F \colon 2^{[n]} \to \mathbb{R}$ is called *submodular* if $F(A) + F(B) \geq F(A \cup B) + F(A \cap B)$ for all $A, B \subseteq [n]$. $F$ is called *modular* if equality holds for all $A, B \subseteq [n]$.

Since a set function $F$ is submodular if and only if its Lovász extension $f = \Phi^{-1}(F)$ is convex (Lovász (1983)), we can specialize Problem 1.1 in the setting of this section as follows.

**Problem 5.3.** *Given a set function $F \in \mathcal{F}_n$, how to decompose it into a difference of submodular set functions such that their Lovász extensions have as few pieces as possible?*

Having a Lovász extension with few pieces is desirable because it allows the submodular function to be stored and accessed efficiently during computational tasks. Moreover, as we argue in Appendix D, for accordingly normalized submodular functions, the number of pieces of the Lovász extension is precisely the number of vertices of the base polytope, which, in turn, is precisely the Newton polytope of the Lovász extension.

As Problem 5.3 is a special case of Problem 1.1, we can translate our results from Section 3 to the setting of submodular functions.

**Corollary 5.4** (informal)**.** *The set of decompositions of a general set function into a difference of submodular functions (modulo modular functions) is a polyhedron that arises as the intersection of two shifted copies of the cone of submodular functions. In analogy to our general results, the* irreducible *decompositions correspond precisely to the bounded faces of that polyhedron and every* minimal *decomposition is a vertex.*

Example D.14 shows that the Lovász extensions of cut functions are hyperplane functions and thus admit a unique minimal decomposition into submodular functions, which are themselves cut functions. In particular, the Lovász extensions of the decomposition have at most as many pieces as the Lovász extensions of the original cut function.

## 6 Neural Network Constructions

In this section we consider the following question: Given a CPWL function $f\colon \mathbb{R}^n \to \mathbb{R}$ with $q$ pieces and $k$ affine components, what is the necessary depth, width, and size of a neural network exactly representing this function? To this end, we first discuss the necessary background on neural networks and known results on neural complexity. We then prove better results for the case of $f$ being convex. Finally, we extend these results to nonconvex functions by writing them as a difference of convex functions.

**Background.** For a number of hidden layers $d \geq 0$, a *neural network* with *rectified linear unit* (ReLU) activiations is defined by a sequence of $d+1$ affine transformations $T_i\colon \mathbb{R}^{n_{i-1}} \to \mathbb{R}^{n_i}$, $i \in [d+1]$. We assume that $n_0 = n$ and $n_{d+1} = 1$. If $\sigma$ denotes the function that computes the ReLU function $x \mapsto \max\{x, 0\}$ in each component, the neural network is said to compute the function $f\colon \mathbb{R}^n \to \mathbb{R}$ given by $f = T_{d+1} \circ \sigma \circ T_d \circ \sigma \circ \cdots \circ \sigma \circ T_1$. We say that the neural network has *depth* $d + 1$, *width* $\max_{i \in [d]} n_i$, and *size* $\sum_{i \in [d]} n_i$.

It is well-known that the maximum of $n$ numbers can be computed with depth $\lceil \log_2 n \rceil + 1$ and overall size $\mathcal{O}(n)$ (Arora et al., 2018). This simple fact has been used in the literature to deduce exact representations of CPWL functions with neural networks from known representations of CPWL functions. We would like to focus on two of them here, which are in a sense incomparable. The first one goes back to Hertrich et al. (2021) and builds upon ideas from Wang & Sun (2005). We present it here in a slightly stronger form.

**Theorem 6.1.** *Every CPWL function $f\colon \mathbb{R}^n \to \mathbb{R}$ with $k$ affine components can be represented by a neural network with depth $\lceil \log_2(n + 1) \rceil + 1$ and overall size $\mathcal{O}(k^{n+1})$.*

The second one goes back to Chen et al. (2022) and is based on the lattice representation of CPWL functions, compare Tarela & Martinez (1999).

**Theorem 6.2** ((Chen et al., 2022))**.** *Every CPWL function $f\colon \mathbb{R}^n \to \mathbb{R}$ with $q$ pieces and $k$ affine components can be represented by a neural network with depth $\lceil \log_2 p \rceil + \lceil \log_2 q \rceil + 1$ and overall size $\mathcal{O}(kq)$.*

As noted before, one can assume that $n < k \leq q$, since otherwise we could affinely project to a lower dimension without losing information. In fact, one would usually assume that the input dimension $n$ is much lower than the number of affine components $k$. Therefore, Theorem 6.1 provides the better representation in terms of depth, while Theorem 6.2 provides the better representation in terms of size. However, both theorems are kind of inflexible for the user, dictating a certain depth and providing only these two specific options. So, the naturally occurring question is: can we somehow freely choose a depth and trade depth against size in these representations? In other words: can we smoothly interpolate between the low-depth high-size representation of Theorem 6.1 and the low-size high-depth representation of Theorem 6.2? In the remaining section we present results that achieve this to some extent.

**New Constructions for the Convex Case.** In this part we prove that we can achieve the desired tradeoff easily in the convex case by mixing the two representations of Theorem 6.1 and Theorem 6.2.

**Theorem 6.3.** *Every convex CPWL function $f\colon \mathbb{R}^n \to \mathbb{R}$ with $k$ affine components can be represented by a neural network with depth $\lceil \log_2(n + 1) \rceil + \lceil \log_2 r \rceil + 1$ and overall size $\mathcal{O}(rs^{n+1})$, for any free choice of parameters $r$ and $s$ with $rs \geq k = q$.*

In order to see how this provides a tradeoff between the representations of Theorem 6.1 and Theorem 6.2, it is worth looking at the extreme cases. If we choose $r = 1$ and $s = k$, we exactly obtain the bounds from Theorem 6.1. On the other hand, if we choose $r = k$ and $s = 1$, we obtain a neural network with depth $\mathcal{O}(\log k)$ and size $\mathcal{O}(k)$, which is qualitatively close to the bounds of Theorem 6.2. In fact, the even better size bound stems from the fact that our construction heavily relies on convexity.

In conclusion, by choosing an appropriate $r$ (and corresponding $s$), the user can freely decide how to trade depth against size in neural network representations and thereby interpolate between the two extreme representations of Theorem 6.1 and Theorem 6.2.

**Extension to the Nonconvex Case.** The construction of Theorem 6.3 provides a nice blueprint of how to interpolate between Theorem 6.1 and Theorem 6.2, but it has the big limitation that it only works in the convex case. Simply mixing the two known representations of Theorems 6.1 and 6.2 does not appear to work in the nonconvex case, as one cannot as easily identify groups of affine components that can be treated separately.

Instead we propose a different approach: given a CPWL function, first split it into a difference of two convex ones and then apply Theorem 6.3 to these two functions. To do this efficiently, it requires to find a good answer to Problem 1.1.

As discussed in this paper, it is quite challenging to give a satisfying answer to Problem 1.1 in full generality, but there are special cases, where we do have a good answer; see Section 3. For example, we obtain the following result by combining Theorem 6.3 with Lemma E.3.

**Corollary 6.4.** *Let $f \colon \mathbb{R}^n \to \mathbb{R}$ be a CPWL function that is compatible with a* regular *polyhedral complex $\mathcal{P}$ with $\tilde{q} = |\mathcal{P}^n|$ full-dimensional polyhedra. Then, $f$ can be represented by a neural network with depth $\lceil \log_2(n+1) \rceil + \lceil \log_2 r \rceil + 1$ and overall size $\mathcal{O}(rs^{n+1})$, for any free choice of parameters $r$ and $s$ with $rs \geq \tilde{q}$.*

For a given CPWL function $f$, provided that one can find a regular polyhedral complex such that $\tilde{q}$ is not much larger than the number of pieces $q$, Corollary 6.4 does indeed provide a smooth tradeoff between Theorem 6.1 and Theorem 6.2 in the general, nonconvex case. As $q$ is the minimal number of fulldimensional polyhedra in *any* (potentially nonregular) polyhedral complex compatible with $f$, the big question is how much of a restriction the assumption of regularity might be.

## 7 Open Problems

From the theoretical perspective, the maybe most dominant open question is the following precise version of Problem 1.1.

**Problem 7.1.** *Given a CPWL function $f$ in dimension $n$ with $q$ pieces, does there always exist a decomposition $f = g - h$ such that the number of pieces of $g$ and $h$ is polynomial in $n$ and $q$?*

Note that by definition, $f$ is compatible with a polyhedral complex $\mathcal{P}$ with $|\mathcal{P}^n| = q$. To answer Problem 7.1 positively, by Lemma E.3 it would be sufficient to find a *regular* polyhedral complex $\mathcal{Q}$ with $|\mathcal{Q}^n| = \text{poly}(n, q)$ that is a refinement of $\mathcal{P}$. Conversely, if we can answer Problem 7.1 positively, then the underlying complex of $g + h$ would give us such a regular complex $\mathcal{Q}$. Therefore, to solve Problem 7.1, one needs to answer the following question: given an arbitrary complete polyhedral complex $\mathcal{P}$, what is the "coarsest" refinement of $\mathcal{P}$ that is regular? A positive answer to Problem 7.1 would have useful consequences for our two applications in the context of (submodular) set function optimization and neural network representations. However, also a negative answer would be equally interesting.

One possible approach to resolve Problem 7.1 could be to analyze which objective direction according to Theorem 3.18 leads to a good vertex of the decomposition polyhedron and prove theoretical properties about that vertex. The same theorem might also be key to developing algorithms that find good decompositions with linear programming. Generally, while beyond the scope of this paper, turning any of our insights into practical algorithms, preferably with theoretical guarantees, is a broad avenue for future research.

ACKNOWLEDGMENTS

Marie-Charlotte Brandenburg was partially supported by the Wallenberg AI, Autonomous Systems and Software Program (WASP) funded by the Knut and Alice Wallenberg Foundation. Moritz Grillo is supported by the Deutsche Forschungsgemeinschaft (DFG, German Research Foundation) under Germany's Excellence Strategy — The Berlin Mathematics Research Center MATH+ (EXC-2046/1, project ID: 390685689). Part of this work was completed while Christoph Hertrich was affiliated with Université Libre de Bruxelles, Belgium, and received support by the European Union's Horizon Europe research and innovation program under the Marie Skłodowska-Curie grant agreement No 101153187—NeurExCo.

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

## A  EXAMPLES

### A.1  DIFFERENT PARAMETERIZATIONS OF MEDIAN FUNCTION

**Example A.1** (Median)**.** In this example, we have a look at the different parameterizations for the function that computes the median of 3 numbers. To have a 2-dimensional example, we set $x_3 := 0$. Let the polyhedral complex $\mathcal{P}$ be induced by the maximal polyhedra $P_\pi = \{x \in \mathbb{R}^2 \mid x_{\pi(1)} \le x_{\pi(2)} \le x_{\pi(3)}\}$ where $\pi \colon [3] \to [3]$ is a permutation and $f \colon \mathbb{R}^3 \to \mathbb{R}$ be the function given by $f|_{P_\pi}(x) = x_{\pi(2)}$. The function $f$ has concave breakpoints whenever the median and the higher coordinate change, i.e., at the facets that are given as $\sigma_{\pi,1} = \{x \in \mathbb{R}^2 \mid x_{\pi(1)} \le x_{\pi(2)} = x_{\pi(3)}\}$ and convex breakpoints whenever the median and the lower coordinate change, that is, at the facets that are given as $\sigma_{\pi,2} = \{x \in \mathbb{R}^2 \mid x_{\pi(1)} = x_{\pi(2)} \le x_{\pi(3)}\}$. Since $\|e_1 - e_2\|_2 = \sqrt{2}$ and $\|e_1\|_2 = \|e_2\|_2 = 1$, it holds that

$$w_f(\sigma_{\pi,1}) = \begin{cases} -\sqrt{2} & x_{\pi(1)} = x_3 \\ -1 & x_{\pi(1)} \ne x_3 \end{cases} \quad \text{and} \quad w_f(\sigma_{\pi,2}) = \begin{cases} \sqrt{2} & x_{\pi(3)} = x_3 \\ 1 & x_{\pi(3)} \ne x_3. \end{cases}$$

See Figure 1 for a 2-dimensional illustration.

### A.2  ILLUSTRATION OF DEFINITION 3.12

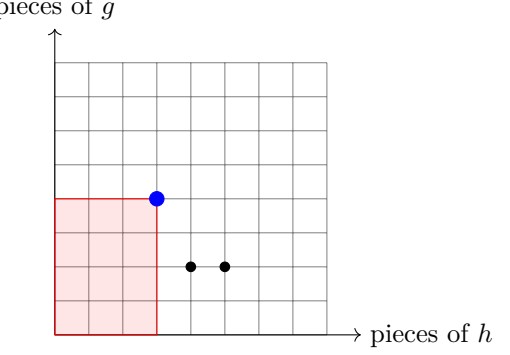
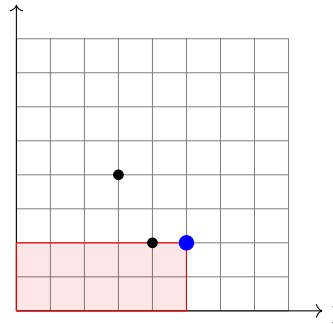

(a) The blue point corresponds to a minimal decomposition.

(b) The blue point corresponds to a decomposition that is not minimal.

Figure 2: Visualization of minimality, where a decomposition $(g, h)$ is described by the number of pieces of $g$ and $h$. A decomposition is minimal, if the rectangle spanned with $(0, 0)$ does not contain another decomposition.

### A.3  EXAMPLES FOR UNIQUE MINIMAL DECOMPOSITIONS

**Example A.2** (Minimal decomposition for hyperplane functions)**.** Let $f \colon \mathbb{R}^n \to \mathbb{R}$ be a hyperplane function given as $f(x) = \sum_{i \in [k]} \lambda_i \cdot \max\{\langle x, a_i \rangle + b_i, \langle x, c_i \rangle + d_i\}$. We can assume without loss of generality that the hyperplanes

$$H_i = \{x \in \mathbb{R}^n \mid \langle x, a_i \rangle + b_i = \langle x, c_i \rangle + d_i\}$$

are pairwise distinct, because otherwise we can simply adjust $\lambda_i$. The polyhedral complex $\mathcal{P}$ induced by the hyperplane arrangement $\{H_i\}_{i \in [k]}$ is compatible with $f$. The convex functions $g, h$ given by

$$g(x) = \sum_{\lambda_i \ge 0} \lambda_i \cdot \max\{\langle x, a_i \rangle + b_i, \langle x, c_i \rangle + d_i\} \text{ and } h(x) = -\sum_{\lambda_i < 0} \lambda_i \cdot \max\{\langle x, a_i \rangle + b_i, \langle x, c_i \rangle + d_i\}$$

are the unique minimal decomposition of $f$ since $\mathrm{supp}_{\mathcal{P}}(g) = \mathrm{supp}_{\mathcal{P}}^+(f) = \{\sigma \in \mathcal{P}^{n-1} \mid \sigma \subseteq \bigcup_{\lambda_i \ge 0} H_i\}$ and $\mathrm{supp}_{\mathcal{P}}(h) = \mathrm{supp}_{\mathcal{P}}^-(f) = \{\sigma \in \mathcal{P}^{n-1} \mid \sigma \subseteq \bigcup_{\lambda_i < 0} H_i\}$.

**Example A.3** (Minimal decomposition of $k$-th order statistic)**.** We construct a polyhedral complex that is compatible with the function $f \colon \mathbb{R}^n \to \mathbb{R}$ that outputs the $k$-th largest entry of $x \in \mathbb{R}^d$. For $U \subseteq [n]$ with $|U| = k$ and $i \in [n] \setminus U$, let

$$P_{i,U} = \{x \in \mathbb{R}^n \mid x_j \leq x_i \leq x_\ell \ \forall \ell \in U, j \in [n] \setminus U\}.$$

All such polyhedra and their faces form a polyhedral complex $\mathcal{P}$ that is compatible with the function $f \colon \mathbb{R}^n \to \mathbb{R}$ given by $f(P_{i,U}) = x_i$. It is not hard to see that $f = g - h$ where $g, h \in \overline{\mathcal{V}}_{\mathcal{P}}^+$ are convex functions given by

$$g(x) := \max_{\substack{I \subseteq [n] \\ |I|=k}} \left( \sum_{i \in I} x_i \right) \ \text{ and } h(x) := \max_{\substack{I \subseteq [n] \\ |I|=k-1}} \left( \sum_{i \in I} x_i \right).$$

Moreover, let $\sigma_{i,j,U} := \{x \in \mathbb{R}^n \mid x_\ell \leq x_j = x_i \leq x_m \text{ for all } m \in U, \ell \in [n] \setminus U\}$. Then $\mathrm{supp}_{\mathcal{P}}(g) = \mathrm{supp}_{\mathcal{P}}^+(f) = \{\sigma_{i,j,U} \mid U \subseteq [n], |U| = k+1\}$ and $\mathrm{supp}_{\mathcal{P}}(h) = \mathrm{supp}_{\mathcal{P}}^-(f) = \{\sigma_{i,j,U} \mid U \subseteq [n], |U| = k\}$. Thus, Proposition 3.15 implies that $(g, h)$ is the unique vertex of every regular polyhedral complex compatible with $f$.

# B Constructions by Kripfganz & Schulze (1987)

**Construction B.1** (Hyperplane extension)**.** For all convex breakpoints, the local convex functions are extended to global convex functions with breakpoints supported on a single hyperplane, and the function $g$ is defined as the sum of all these functions. To analyze it in our framework, let $\mathcal{P}$ be any polyhedral complex that is compatible with $f$ and let $w_f \in \mathcal{W}_{\mathcal{P}}$ be the weight function corresponding to $f$. For $\sigma \in \mathcal{P}^{n-1}$, let $H_\sigma$ be the hyperplane spanned by $\sigma$ and $\mathcal{A}_f^+ = \{H_\sigma \mid w_f(\sigma) > 0\}$ be the hyperplane arrangement consisting of the hyperplanes supporting the breakpoints where $f$ is convex. Let $\mathcal{H}_f^+$ be the common refinement of the polyhedral complex induced by $\mathcal{A}_f^+$ and $\mathcal{P}$. The weight function $w_g \colon \mathcal{P}^{n-1} \to \mathbb{R}$ given by $w_g(\sigma) := \sum_{\substack{\sigma \subseteq H_{\sigma'}, \\ w_f(\sigma')>0}} w_f(\sigma')$ is in $\mathcal{W}_{\mathcal{H}_f^+}$ and nonnegative and hence the corresponding function $g \in \mathcal{V}_{\mathcal{H}_f^+}$ is convex. It follows that $h := g - f$ is convex as well, yielding the desired decomposition.

**Construction B.2** (Local Maxima Decomposition)**.** Let $\{P_1, \ldots, P_m\} = \mathcal{P}^n$ and $f_i$ be the unique linear extension of $f|_{P_i}$. Moreover, let $M_i := \{j \in [m] \mid f_i(x) \geq f_j(x) \text{ for all } x \in P_i\}$ and $g_i := \max_{j \in M_i} P_j$. Then

$$f = \min_{i \in [m]} \max_{j \in M_i} f_i = \min_{i \in [m]} g_i$$

and $g := \sum_{i \in [m]} g_i$ is a convex function. Furthermore, let $h_i := g - g_i$, then $h := \max_{i \in [m]} h_i$ is a convex function as well and it holds that

$$g - h = g - \max_{i \in [m]}(g - g_i) = g - (g - \min_{i \in [m]} g_i) = g - (g - f) = f$$

Let $H_{i,j} := \{x \in \mathbb{R}^n \mid f_i(x) = f_j(x)\}$ and $\mathcal{A}_f = \{H_{i,j} \mid i \neq j\}$. Furthermore, let $\mathcal{H}_f$ be the polyhedral complex induced by the hyperplane arrangement $\mathcal{A}_f$. Then we have that $g, h \in \mathcal{V}_{\mathcal{H}_f}$.

**Proposition B.3.** *There is a CPWL function $f$ and convex CPWL functions $g, h$ with $f = g - h$ such that every decomposition $(g', h') \in \mathcal{D}_{\mathcal{H}_f}(f)$ as well as every decomposition $(g', h') \in \mathcal{D}_{\mathcal{H}_f^+}(f)$ is dominated by $(g, h)$.*

*Proof.* Let $\mathcal{P}$ be the polyhedral complex in $\mathbb{R}^2$ with rays $\rho_1 = \mathrm{cone}((1,0))$, $\rho_2 = \mathrm{cone}((0,1))$, $\rho_3 = \mathrm{cone}((1,2))$ and $\rho_4 = \mathrm{cone}((2,1))$. Let $w_f(\rho_1) = w_f(\rho_2) = 1$ and $w_f(\rho_3) = w_f(\rho_4) = \frac{\sqrt{5}}{3}$. Then according to Theorem C.1 the unique minimal decomposition is given by the complex obtained by adding the ray $\rho_5 = \mathrm{cone}((-1,-1))$ and the weight function

$$w_g(\rho) = \begin{cases} w_f(\rho) & \rho \in \mathrm{supp}_{\mathcal{P}}^+(f) \\ 0 & \rho \in \mathrm{supp}_{\mathcal{P}}^-(f) \\ \sqrt{2} & \rho = \rho_5 \end{cases}$$

as well as $w_h = w_g - w_f$. Nevertheless, the ray $\rho_5$ is not contained in (the support of) $\mathcal{H}_f$ and hence this solution is not in $\mathcal{D}_{\mathcal{H}_f}(f)$. Since $(g, h)$ is the unique (up to adding a linear function) minimal decomposition, it follows that any solution in $\mathcal{D}_{\mathcal{H}_f}(f)$ must be dominated by $(g, h)$. Since $\mathcal{H}_f^+$ is a coarsening of $\mathcal{H}_f$, it holds that every decomposition in $\mathcal{D}_{\mathcal{H}_f^+}(f)$ is contained in $\mathcal{D}_{\mathcal{H}_f}(f)$ as well, impyling the result for $\mathcal{D}_{\mathcal{H}_f^+}(f)$. $\qquad\square$

### B.1 Examples of existing decompositions

**Example B.4** (hyperplane extension of $k$-th order statistic). Let $f$ be the function from Example A.3. For any $i, j \in [n]$ and $U \subseteq [n]$ with $i, j \in U$ and $|U| = k + 1$ it holds that $\sigma_{i,j,U} \in \mathrm{supp}^+(f)$ and $H_{\sigma_{i,j,U}} = \{x \in \mathbb{R}^n \mid x_i = x_j\}$. Hence, $\mathcal{P}_g$ is the braid fan and it holds that $g(x) = \binom{n}{k-1} \sum_{i \neq j} \max\{x_i, x_j\}$. Thus, the unique vertex $(g^*, h^*)$ from Example A.3 is clearly a non-trivial coarsening of the decomposition obtained from the hyperplane extension (since $g^*$ is a non-trivial coarsening of $g$) and hence the decomposition cannot be a vertex of $\mathcal{D}_{\mathcal{Q}}(f)$ for any regular polyhedral complex $\mathcal{Q}$.

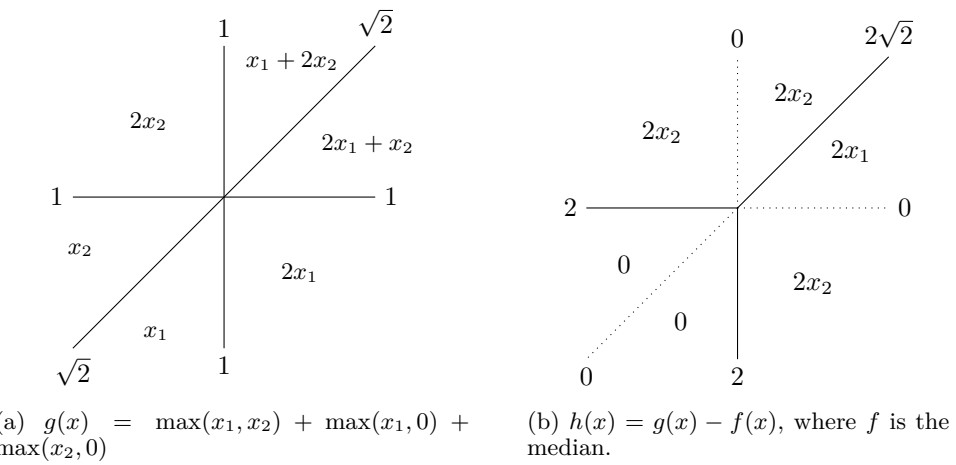

(a) $g(x) = \max(x_1, x_2) + \max(x_1, 0) + \max(x_2, 0)$

(b) $h(x) = g(x) - f(x)$, where $f$ is the median.

Figure 3: The hyperplane extension of the median (second largest number) of $0, x_1, x_2$ (i.e., $n = 3$) (Example B.4), which agrees with the local maxima decomposition (Example B.5) up to a factor 2. These representations do not agree for the median when $n > 3$.

**Example B.5** (local maxima decomposition of $k$-th order statistic). Let $f$ be the function from Example A.3. Then, for $U \subseteq [n]$ with $|U| = k - 1$ and $i \in [n] \setminus U$, we have that $g_{i,U}(x) = \max_{j \in [n] \setminus U} x_j$. Thus,

$$g(x) = \sum_{i,U} g_{i,U}(x) = (n - k + 1) \cdot \sum_{\substack{S \subseteq [n] \\ |S| = n-k+1}} \max_{j \in S} x_j.$$

Note that $g$ has only breakpoints when two coordinates that are the two highest coordinates in some set $S$ swap places in the ordering. So, for any $T \subseteq [n]$ such that $|T| = n - k$ and any bijection $\pi \colon [k] \to [n] \setminus T$, let

$$P_{T,\pi} := \{x \in \mathbb{R}^n \mid x_j \leq x_{\pi(1)} \leq \dots \leq x_{\pi(k+1)}, j \in T\}.$$

It follows that the set of full-dimensional cones $\mathcal{P}_g^n$ of the unique coarsest polyhedral complex $\mathcal{P}_g$ compatible with $g$ is given as $\mathcal{P}_g^n = \{P_{\pi,T}\}_{\pi,T}$. Again, the unique vertex $(g^*, h^*)$ from Example A.3 is clearly a non-trivial coarsening of the decomposition obtained from the lattice representation and hence the decomposition cannot be a vertex of $\mathcal{D}_{\mathcal{Q}}(f)$ for any regular polyhedral complex $\mathcal{Q}$.

## C    Counterexample to a construction of Tran & Wang (2024)

### C.1    Duality and Newton polytopes

In this section we describe the duality between convex piecewiese linear functions and Newton polytopes, adapted to our setup.

A *positively homogenous* convex CPWL function is a function $f$ such that $f(0) = 0$, and $\mathcal{P}_f$ is a polyhedral fan. In this case, it can be written as $f(x) = \max_{i \in [k]} \langle x, v_i \rangle$, where $v_i \in \mathbb{R}^n$. We define the *Newton polytope* of $f$ as the convex hull $\mathrm{Newt}(f) = \mathrm{conv}(v_1, \ldots, v_k)$. Then $f$ is the *support function* of $\mathrm{Newt}(f)$, i.e., $f(x) = \max_{p \in \mathrm{Newt}(f)} \langle p, x \rangle$. We now give an interpretation of $\mathcal{P}_f$ and $w_f$ in terms of the Newton polytope.

Given any $n$-dimensional polytope $P \subset \mathbb{R}^n$, the (outer) normal cone of a $k$-dimensional face $F$ of $P$ is the $(n-k)$-dimensional cone

$$N_F(P) = \left\{ x \in \mathbb{R}^n \;\middle|\; \langle z, x \rangle = \max_{p \in P} \langle p, x \rangle \text{ for all } z \in F \right\}. \tag{1}$$

In particular, if $P = \mathrm{Newt}(f)$ and $v$ is a vertex of $P$, then the description of the normal cone turns into

$$N_v(\mathrm{Newt}(f)) = \{ x \in \mathbb{R}^n \mid \langle v, x \rangle = f(x) \},$$

and agrees with a maximal polyhedron in $\mathcal{P}_f^n$. The *normal fan* of a polytope is the collection of normal cones over all faces. Thus, for positively homogeneous convex functions, the polyhedral complex $\mathcal{P}_f$ agrees with the normal fan of $\mathrm{Newt}(f)$, and the number of linear pieces of $f$ equals the number of vertices of $\mathrm{Newt}(f)$. The duality between $\mathcal{P}_f$ and $\mathrm{Newt}(f)$ also establishes a bijection between faces $\sigma \in \mathcal{P}_f^{n-1}$ and edges of $\mathrm{Newt}(f)$, and for the corresponding weight function $w_f \in \mathcal{W}_\mathcal{P}$ holds that $w_f(\sigma)$ equals the Euclidean length of the edge that is dual to $\sigma$. This correspondence extends to general convex CWPL functions, where $\mathcal{P}_f$ is a complex which is dual to a polyhedral subdivision of $\mathrm{Newt}(f)$, and $w_f$ corresponds to lengths of edges in this subdivision (Maclagan & Sturmfels, 2015, Chapter 3.4).

### C.2    The Construction from Tran & Wang (2024)

The duality between positively homogeneous convex CPWL functions and Newton polytopes, as described in Appendix C.1, serves as a motivation for Tran & Wang (2024) to construct minimal decompositions $f = g - h$ of positively homogeneous CPWL functions as the difference of two convex such functions in dimension 2.

**Theorem C.1** (Tran & Wang (2024)). *For every positively homogeneous CPWL-function $f \colon \mathbb{R}^2 \to \mathbb{R}$ exists a unique (up to adding a linear function) minimal representation as difference of two convex functions $g, h$.*

The decomposition can be obtained as follows. Let $\mathcal{P}_f$ be a 2-dimensional polyhedral fan compatible with $f$ with rays $\rho_1, \ldots, \rho_m \subset \mathbb{R}^2$ and ray generators $r_1, \ldots, r_m \in \mathbb{R}^2$ such that $\|r_i\| = 1$ for $i = 1, \ldots, m$. Furthermore, let $w_f$ be the corresponding element in $\mathcal{W}_{\mathcal{P}_f}$ and $w_f^+ := \max\{w_f, 0\}$. We now define an additional ray $\rho_{m+1}$ with ray generator $r_{m+1} = -\sum_{i=1}^m \max(w_f(\rho_i), 0) r_i$ and a convex function $g$ through the weights

$$w_g(\rho_i) = \begin{cases} w_f(\rho_i) & \text{if } w_f(\rho_i) > 0, i \in [m] \\ 0 & \text{if } w_f(\rho_i) \leq 0, i \in [m] \\ \sum_{i=1}^m \max(w_f(\rho_i), 0) & \text{if } i = m + 1. \end{cases}$$

This defines the convex functions $g$, $h = g - f$, and results in a minimal decomposition $f = g - h$ in the 2-dimensional positively homogeneous case. Considering this construction through to the duality to Newton polytopes, we can identify rays of $\mathcal{P}_f$ which correspond to convex breakpoints of $f$ with edges of the Newton polytope $\mathrm{Newt}(g)$, and the construction from Theorem C.1 adds a "missing" edge to the Newton polygon $\mathrm{Newt}(g)$. We now describe the proposed construction to generalize the 2-dimensional method to higher dimensions.

**Construction C.2** (Tran & Wang (2024, Section 4.1)). Let $f\colon \mathbb{R}^n \to \mathbb{R}$ be a positively homogeneous CWPL-function and $\mathcal{P}$ a polyhedral fan compatible with $f$. The attempt is to balance $w_f^+$ locally around every $\tau \in \mathcal{P}^{n-2}$ and then "glue together" the local balancings to a global balancing. So, for some $\tau \in \mathcal{P}^{n-2}$, suppose that $\{\sigma_1, \dots \sigma_k\} = \mathrm{star}_{\mathcal{P}}(\tau)$ are the cones containing $\tau$. The rays spanned by $e_{\sigma_i/\tau}$ that inherit the weights $w_f^+(\sigma_i)$ for $i \in [k]$ induce a 2-dimensional fan $\mathcal{P}_\tau$ in the 2-dimensional linear space $\mathrm{span}(\tau)^\perp$ orthogonal to $\mathrm{span}(\tau)$. Let $P_\tau$ be the polygon in $\mathrm{span}(\tau)^\perp$ corresponding to the minimal balancing of $w_f^+$ regarded as map $w_f^+\colon \mathcal{P}_\tau^1 \to \mathbb{R}$. Now proceed with the following steps.

1. For every $\tau \in \mathcal{P}^{n-2}$, construct the polygon $P_\tau$.

2. Place the polygons $P_\tau$ in $\mathbb{R}^n$ in such a way, that whenever $\tau_1, \tau_2 \in \mathcal{P}^{n-2}$ are faces of $\sigma \in \mathrm{supp}_{\mathcal{P}}^+(f)$, then the edges in $P_{\tau_1}$ and $P_{\tau_2}$ that correspond to $\sigma$ are identified with each other.

3. Take the convex hull $P_g$ of the polygons $\{P_\tau\}_{\tau \in \mathcal{P}^{n-2}}$.

4. The support function $g$ of the polytope $P_g$ and $h \coloneqq g - f$ are a decomposition of $f$.

One can check that for some $\sigma \in \mathcal{P}^{n-1}$ and $\tau \in \mathcal{P}^{n-2}$ being a face of $\sigma$, the edge $e_\sigma$ of length $w(\sigma)$ which is perpendicular in $\mathrm{span}(\tau)^\perp$ to $\tau$ is independent of the choice of the face $\tau$. In particular, the direction of the edge $e_\sigma$ is normal to the hyperplane spanned by $\sigma$. However, it remained unclear, whether or not, the second step in this procedure is always well-defined, that is, that placing the polygons in such a coherent way is possible. To make this more precise, let for some $\tau \in \mathcal{P}^{n-2}$ the edges of the polygon $P_\tau$ be given in a cyclic way $\{e_{\sigma_1}, \dots e_{\sigma_m}\}$. Placing a polygon $P_\tau$ refers to choosing an $x_\tau \in \mathbb{R}^n$ and defining the placed polygon as $P_\tau(x_\tau) = \mathrm{conv}(x_\tau, x_\tau + e_{\sigma_1}, x_\tau + e_{\sigma_1} + e_{\sigma_2}, \dots, x_\tau + \sum_{i=1}^m e_{\sigma_m})$. Placing them in a coherent way means choosing an $x_\tau \in \mathbb{R}^n$ for every $\tau \in \mathcal{P}^{n-2}$ such that $P_{\tau_1}(x_{\tau_1}) \cap P_{\tau_2}(x_{\tau_2}) = \mathrm{conv}(x_\sigma, x_\sigma + e_\sigma)$ for some $x_\sigma \in \mathbb{R}^n$ whenever $\tau_1$ and $\tau_2$ are faces of $\sigma$. A priori it is not clear that such $x_\tau$ always exist. The following example will in fact show that the resulting linear equation system not always yields a solution.

## C.3 Counterexample to the construction

In the remaining of this section, we give a counterexample to Construction C.2, which is stated in Tran & Wang (2024) in terms of (virtual) Newton polytopes as a potential generalization of the 2-dimensional construction to higher dimensions.

**Example C.3** (Counterexample to Construction C.2). Figure 4 is an illustration of 4 polygons with labelled edges that cannot be placed in $\mathbb{R}^3$ such that the edges of different polygons with the same label are identified with each other. Hence, applying the above procedure to the CPWL-function $f\colon \mathbb{R}^3 \to \mathbb{R}$ given by

$$f(x) = \max\{0, \max_{\substack{i,j \in [3] \\ i \neq j}} \{\min\{x_i, x_j - x_i\}\}\}$$

is not well-defined since these 4 polygons arise and should be identified in the indicated way, which is impossible.

We describe the 2-skeleton of a polyhedral fan $\mathcal{P}$ that is compatible with $f$. Let $e_i$ be the $i$-th standard unit vector. The rays are given as follows:

$$\mathcal{P}^1 = \{\mathrm{cone}(-e_i), \mathrm{cone}(e_i), \mathrm{cone}(e_i + e_j), \mathrm{cone}(e_i + e_j + 2e_k), \mathrm{cone}(e_i + 2e_j + 2e_k),$$
$$\mathrm{cone}(e_i + e_j + 2e_k), \mathrm{cone}(e_i + e_j + e_k) \mid i, j, k \in [3] \text{ pairwise distinct}\}$$

and the 2-dimensional cones as

$$\mathcal{P}^2 = \{\, \mathrm{cone}(e_i, -e_j), \mathrm{cone}(-e_i, e_i + e_j + 2e_k), \mathrm{cone}(-e_i, e_j + e_k), \mathrm{cone}(e_i, e_i + e_j + 2e_k),$$
$$\mathrm{cone}(e_j + e_k, e_i + e_j + 2e_k), \mathrm{cone}(e_i + e_j + 2e_k, e_i + 2e_j + 2e_k), \mathrm{cone}(e_i + e_j + 2e_k, e_i + e_j + e_k),$$
$$\mathrm{cone}(e_i + 2e_j + 2e_k, e_i + e_j + e_k), \mathrm{cone}(e_j + e_k, e_i + 2e_j + 2e_k) \mid i, j, k \in [3] \text{ pairwise distinct}\}$$

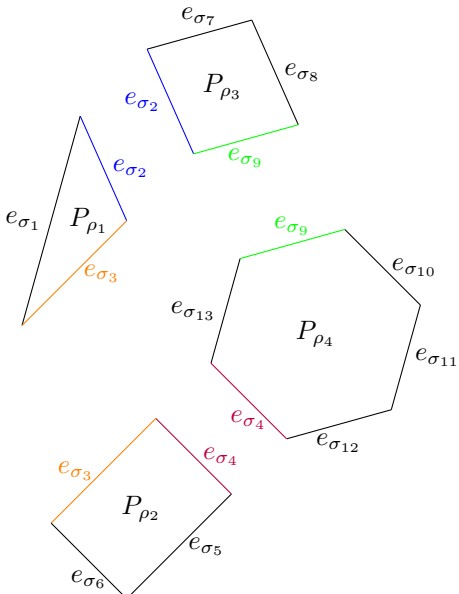

Figure 4: 4 polygons that cannot be placed in a coherent way in $\mathbb{R}^3$

The weight function $w_f \colon \mathcal{P}^2 \to \mathbb{R}$ given by

$$
\begin{aligned}
w(\mathrm{cone}(e_i, -e_j)) &= 1 \\
w(\mathrm{cone}(-e_i, e_i + e_j + 2e_k)) &= -\sqrt{5} \\
w(\mathrm{cone}(-e_i, e_j + e_k)) &= \sqrt{8} \\
w(\mathrm{cone}(e_i, e_i + e_j + 2e_k)) &= \sqrt{2}, \\
w(\mathrm{cone}(e_j + e_k, e_i + e_j + 2e_k)) &= \sqrt{3}, \\
w(\mathrm{cone}(e_i + e_j + 2e_k, e_i + 2e_j + 2e_k)) &= -\sqrt{5}, \\
w(\mathrm{cone}(e_i + e_j + 2e_k, e_i + e_j + e_k)) &= \sqrt{2}, \\
w(\mathrm{cone}(e_i + 2e_j + 2e_k, e_i + e_j + e_k)) &= \sqrt{2}, \\
w(\mathrm{cone}(e_j + e_k, e_i + 2e_j + 2e_k)) &= 0
\end{aligned}
$$

corresponds to the function $f$ and is therefore balanced. See Figure 5 for a 2-dimensional illustration of $\mathcal{P}$.

Consider the 4 rays

$$
\rho_1 = \mathrm{cone}((0,1,1)), \rho_2 = \mathrm{cone}((1,1,2)), \rho_3 = \mathrm{cone}((1,2,1)), \rho_4 = \mathrm{cone}((1,1,1))
$$

We will see that the corresponding polygons $P_{\rho_1}, P_{\rho_3}, P_{\rho_3}$ and $P_{\rho_4}$ equal the ones in Figure 4. The 2-dimensional cones which are in the stars of the 4 rays are the following:

$\sigma_1 = \mathrm{cone}((0,1,1),(-1,0,0))$, $\sigma_2 = \mathrm{cone}((0,1,1),(1,2,1))$, $\sigma_3 = \mathrm{cone}((0,1,1),(1,1,2))$,
$\sigma_4 = \mathrm{cone}((1,1,2),(1,1,1))$, $\sigma_5 = \mathrm{cone}((1,1,2),(1,0,1))$, $\sigma_6 = \mathrm{cone}((1,1,2),(0,0,1))$,
$\sigma_7 = \mathrm{cone}((1,2,1),(0,1,0))$, $\sigma_8 = \mathrm{cone}((1,2,1),(1,1,0))$, $\sigma_9 = \mathrm{cone}((1,2,1),(1,1,1))$,
$\sigma_{10} = \mathrm{cone}((1,1,1),(2,2,1))$, $\sigma_{11} = \mathrm{cone}((1,1,1),(2,1,1)$, $\sigma_{12} = \mathrm{cone}((1,1,1),(2,1,2)$,
$\sigma_{13} = \mathrm{cone}((1,2,2),(1,1,1))$,

The direction of the edges of the polygons are given by the normal vectors of the hyperplanes spanned by the corresponding 2-dimensional cone and their length by the weight of the corresponding 2-dimensional cone.

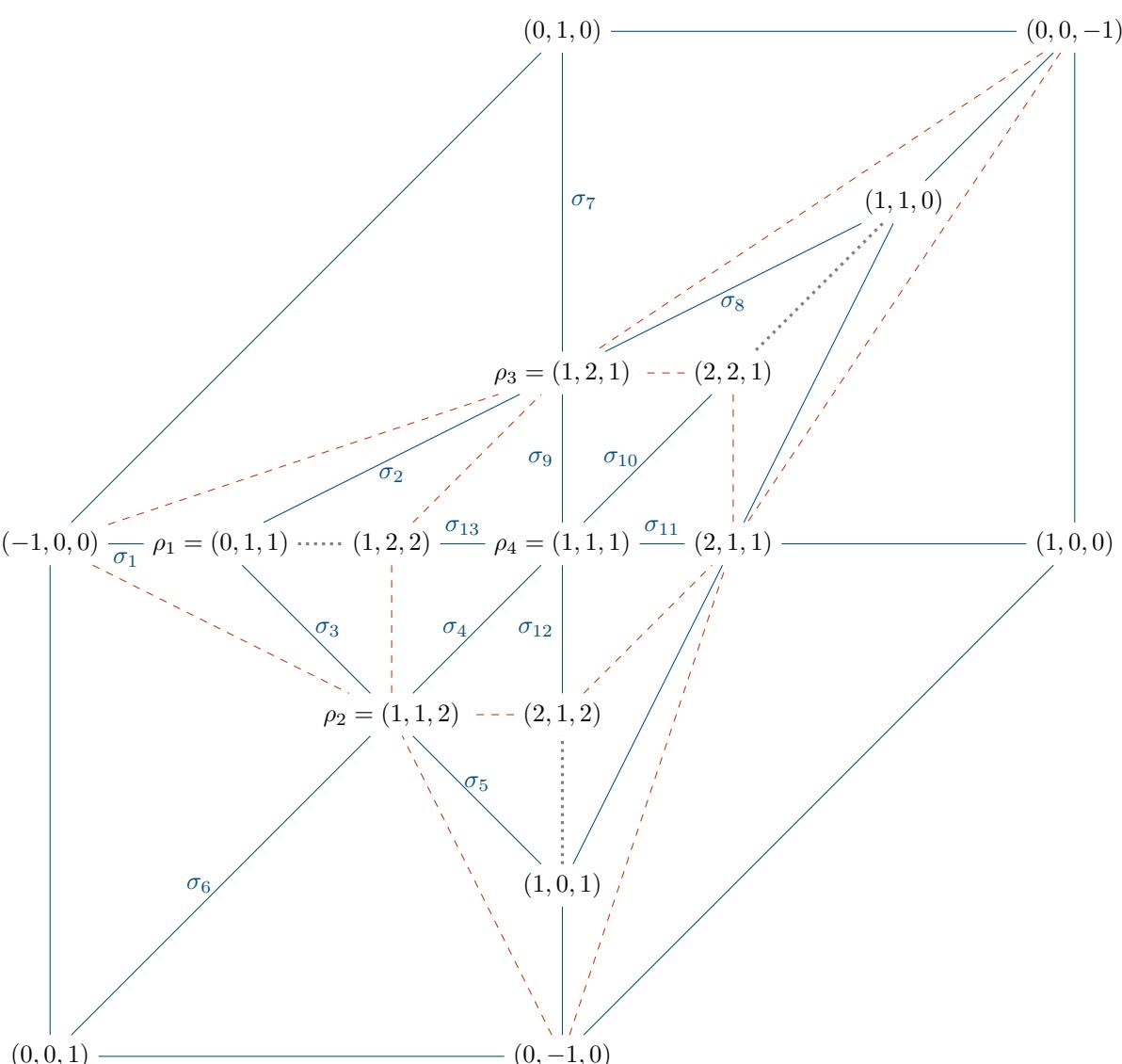

Figure 5: A 2-dimensional representation of $\mathcal{P}$. The blue lines correspond to convex break-points of the function $f$, that is, a cone $\sigma \in \mathcal{P}^2$ such that $w(\sigma) > 0$. The concave breakpoints $(w(\sigma) < 0)$ are dashed and colored in orange. $f$ has no breakpoints on the gray, dotted lines $(w(\sigma) = 0)$.

$$e_{\sigma_1} = (0, 2, -2),\ e_{\sigma_2} = (1, -1, 1),\ e_{\sigma_3} = (-1, -1, 1),\ e_{\sigma_4} = (1, -1, 0),\ e_{\sigma_5} = (1, 1, -1),$$
$$e_{\sigma_6} = (-1, 1, 0),\ e_{\sigma_7} = (1, 0, -1),\ e_{\sigma_8} = (-1, 1, -1),\ e_{\sigma_9} = (-1, 0, 1),\ e_{\sigma_{10}} = (1, -1, 0),$$
$$e_{\sigma_{11}} = (0, -1, 1),\ e_{\sigma_{12}} = (-1, 0, 1),\ e_{\sigma_{13}} = (0, 1, -1)$$

In order to construct the polygons one needs to consider the orientation of the edge (and the normal vector) in the particular polygon. One can convince themselves that the polygons $P_{\rho_1}, P_{\rho_3}, P_{\rho_3}$ and $P_{\rho_4}$ of the 4 rays are given as:

$$P_{\rho_1}(x_{\rho_1}) = \mathrm{conv}(x_{\rho_1} + e_{\sigma_1}, x_{\rho_1} + e_{\sigma_1} + e_{\sigma_2}, x_{\rho_1} + e_{\sigma_1} + e_{\sigma_2} + e_{\sigma_3})$$
$$P_{\rho_2}(x_{\rho_2}) = \mathrm{conv}(x_{\rho_2} - e_{\sigma_3}, x_{\rho_2} - e_{\sigma_3} - e_{\sigma_4}, x_{\rho_2} - e_{\sigma_3} - e_{\sigma_4} - e_{\sigma_5}, x_{\rho_2} - e_{\sigma_3} - e_{\sigma_4} - e_{\sigma_5} - e_{\sigma_6})$$
$$P_{\rho_3}(x_{\rho_3}) = \mathrm{conv}(x_{\rho_3} - e_{\sigma_2}, x_{\rho_3} - e_{\sigma_2} + e_{\sigma_7}, x_{\rho_3} - e_{\sigma_2} + e_{\sigma_7} + e_{\sigma_8}, x_{\rho_3} - e_{\sigma_2} + e_{\sigma_7} + e_{\sigma_8} + e_{\sigma_9})$$
$$P_{\rho_4}(x_{\rho_4}) = \mathrm{conv}(x_{\rho_3} + e_{\sigma_9}, x_{\rho_4} + e_{\sigma_9} + e_{\sigma_{10}}, x_{\rho_4} + e_{\sigma_9} + e_{\sigma_{10}} + e_{\sigma_{11}}, x_{\rho_4} + e_{\sigma_9} + e_{\sigma_{10}} + e_{\sigma_{11}} + e_{\sigma_{12}},$$
$$x_{\rho_4} + e_{\sigma_9} + e_{\sigma_{10}} + e_{\sigma_{11}} + e_{\sigma_{12}} + e_{\sigma_4}, x_{\rho_4} + e_{\sigma_9} + e_{\sigma_{10}} + e_{\sigma_{11}} + e_{\sigma_{12}} + e_{\sigma_4} + e_{\sigma_{13}})$$

Figure 4 already shows that they cannot be placed in a coherent way. To make this mathematically precise, one obtains a system of linear equations for $x_{\rho_1}, x_{\rho_2}, x_{\rho_3}$ and $x_{\rho_4}$, by plugging in the values for the normal vectors, that ensures that the edges for the same 2-dimensional cone in different polygons are identified. This linear equation system does not have a solution.

## D SUBMODULAR FUNCTIONS

This section is a detailed version of Section 5, where we demonstrate that a special case of our framework is to decompose a general set function into a difference of submodular set funtions and translate our results to this setting. Such decompositions are a popular approach to solve optimization problems as disussed in the introduction.

**Definition D.1.** The *braid arrangement* in $\mathbb{R}^n$ is the hyperplane arrangement consisting of the $\binom{n}{2}$ hyperplanes $x_i = x_j$, with $1 \le i < j \le n$.

For the remaining section, let $\mathcal{P}$ be the polyhedral complex arising from the braid arrangement. Let $\mathcal{F}_n$ be the vector space of set functions from $2^{[n]}$ to $\mathbb{R}$. We first show that functions in $\mathcal{V}_{\mathcal{P}}$ are in one-to-one correspondence with the set functions $\mathcal{F}_n$. To this end, for a set $S \subseteq [n]$, let $\mathbb{1}_S = \sum_{i \in S} e_i$ be the indicator vector of $S$, that is, the vector that contains entries 1 for indices in $S$ and 0 otherwise.

**Proposition D.2.** *The mapping $\Phi$ that maps $f \in \mathcal{V}_{\mathcal{P}}$ to the set function $F(S) = f(\mathbb{1}_S)$ is a vector space isomorphism.*

*Proof.* The map $\Phi$ is clearly a linear map. To prove that $\Phi$ is an isomorphism, we show that a function $f \in \mathcal{V}_{\mathcal{P}}$ is uniquely determined by its values on $\{\mathbb{1}_S\}_{S \subseteq [n]}$ and any choice of real values $\{y_S\}_{S \subseteq [n]}$ give rise to a function $f \in \mathcal{V}_{\mathcal{P}}$ such that $f(\mathbb{1}_S) = y_S$.

First, note that the maximal polyhedra $\mathcal{P}^n$ are of the form $P_\pi = \{x \in \mathbb{R}^n \mid x_{\pi(1)} \le \ldots \le x_{\pi(n)}\}$ for a permutation $\pi \colon [n] \to [n]$. There are exactly the $n+1$ indicator vectors $\{\mathbb{1}_{S_i}\}_{i=0,\ldots,n}$ contained in $P_\pi$, where $S_i := \{\pi(n+1-i), \ldots, \pi(n)\}$ for $i \in [n]$ and $S_0 := \emptyset$. Moreover, the vectors $\{\mathbb{1}_{S_i}\}_{i=0,\ldots,n}$ are affinely independent and hence the values $\{f(\mathbb{1}_{S_i})\}_{i=0,\ldots,n}$ uniquely determine the affine linear function $f|_{P_\pi}$. Therefore, $f$ is uniquely determined by $\{f(\mathbb{1}_{S_i})\}_{S \subseteq [n]}$.

Given any values $\{y_S\}_{S \subseteq [n]}$, by the discussion above, there are unique affine linear maps $f|_{P_\pi}$ yielding $f|_{P_\pi}(\mathbb{1}_S) = y_S$ for all $S \subseteq [n]$ such that $\mathbb{1}_S \in P_\pi$. It remains to show that the resulting function $f$ is well-defined on the facets $\mathcal{P}^{n-1}$. Any such facet is of the form

$$\sigma_{\pi,i} = \{x \in \mathbb{R}^n \mid x_{\pi(1)} \le \ldots \le x_{\pi(i)} = x_{\pi(i+1)} \le \ldots \le x_{\pi(n)}\},$$

which is the intersection of $P_\pi$ and $P_{\pi \circ (i,i+1)}$, where $(i, i+1)$ denotes the transposition swapping $i$ and $i+1$. However, the indicator vectors $\{\mathbb{1}_{S_i}\}_{i \in [n] \setminus \{i\}}$ contained in $\sigma_{\pi,i}$ are

a subset of the indicator vectors contained in $P_\pi$ and $P_{\pi \circ (i,i+1)}$. Therefore, it holds that $f|_{P_\pi}(x) = f|_{P_{\pi \circ (i,i+1)}}(x)$ for all $x \in \sigma_{\pi,i}$ implying that $f$ is well-defined as a CPWL function.
□

If we think this the other way around, starting with a set function $F$, then $f = \Phi^{-1}(F)$ is by definition a continuous extension of $F$. It turns out that this particular extension is known as the *Lovász extension* (Lovász, 1983), as we argue below. The Lovász extension is an important concept in the theory and practice of submodular function optimization as it provides a link between *discrete* submodular functions and *continuous* convex functions.

**Definition D.3.** For a set function $F \colon 2^{[n]} \to \mathbb{R}$, the *Lovász extension* $f \colon \mathbb{R}^n \to \mathbb{R}$ is defined by $f(x) = \sum_{i=0}^n \lambda_i F(S_i)$, where $\emptyset = S_0 \subset S_1 \subset \ldots \subset S_n = [n]$ is a chain such that $\sum_{i=1}^n \lambda_i \mathbb{1}_{S_i} = x$ and $\lambda_i \geq 0$ for all $i \in [n-1]$ and $\lambda_0 = 1 - \sum_{i=1}^n \lambda_i$.

**Remark D.4.** In many contexts in the literature, the Lovász extension is only defined on the hypercube $[0,1]^n$. For our purposes, it is more convenient to omit this restriction, which is captured by the above definition.

The intuition of the Lovász extension can already be seen in the proof of Proposition D.2: depending on the ordering of the components of an input vector $x$, the Lovász extension writes $x$ as an affine combination of indicator vectors $\mathbb{1}_{S_i}$, and uses the coefficients of the affine combination to compute the value $f(x)$. Following the intuition, we see in the next proposition that $\Phi^{-1}(F)$ is actually the Lovász extension of $F$.

**Proposition D.5.** *For a set function $F \in \mathcal{F}_n$, the function $f = \Phi^{-1}(F)$ is precisely the Lovász extension of $F$.*

*Proof.* By the definition of the Lovász extension, it follows that it is compatible with $\mathcal{P}$. Thus, the Lovász extension is contained in $\mathcal{V}_\mathcal{P}$. Moreover, as it is an extension that fixes indicator vectors, it follows that $\Phi$ applied to the Lovász extension of $F$ gives us back $F$. As $\Phi$ is an isomorphism by Proposition D.2, the Lovász extension must be exactly $\Phi^{-1}(F)$. □

**Definition D.6.** A set function $F \colon 2^{[n]} \to \mathbb{R}$ is called *submodular* if

$$F(A) + F(B) \geq F(A \cup B) + F(A \cap B) \tag{2}$$

for all $A, B \subseteq [n]$. $F$ is called *modular* if equality holds for all $A, B \subseteq [n]$.

The following well-known property is key to the insights of this section.

**Proposition D.7** (Lovász (1983)). *A set function $F$ is submodular if and only if its Lovász extension $f = \Phi^{-1}(F)$ is convex.*

Applying our insights from Section 3 to the previous proposition, we obtain the following well-known statement.

**Corollary D.8.** *The set of submodular functions forms a polyhedral cone $\mathcal{SM}_n$ in the vector space $\mathcal{F}_n$.*

In particular, we can specialize Problem 1.1 in the setting of this section as follows.

**Problem D.9.** *Given a set function $F \in \mathcal{F}_n$, how to decompose it into a difference of submodular set functions such that their Lovász extensions have as few pieces as possible?*

Having a Lovász extension with few pieces is desirable because it allows the submodular function to be stored and accessed efficiently during computational tasks. As Problem D.9 is a special case of Problem 1.1, we are able to translate our results from Section 3 to the setting of submodular functions.

Let $\mathcal{M}_n \subseteq \mathcal{F}_n$ be the vector space of *modular* functions, that is, set functions that satisfy equation 2 with equality. Note that a set function is modular if and only if its Lovász extension is an affine function (Lovász, 1983). Since for any $M \in \mathcal{M}_n$, a set function $F$ is submodular if and only if $F + M$ is submodular, we define the vector space $\overline{\mathcal{F}}_n = \mathcal{F}_n / \mathcal{M}_n$ of set functions modulo modular functions. Furthermore, let $\overline{\mathcal{SM}}_n$ be the cone of submodular

functions in this quotient. A decomposition $(G, H) \in \overline{\mathcal{SM}}_n \times \overline{\mathcal{SM}}_n$ of a set function $F \in \overline{\mathcal{F}}_n$ is called *irreducible* if there does not exist a submodular function $I \in \overline{\mathcal{SM}}_n \setminus \{0\}$ such that $G - I$ and $H - I$ are submodular. Since a set function $M \in \mathcal{F}_n$ is modular if and only if $\Phi^{-1}(M)$ is affine linear, the isomorphism $\Phi$ of Proposition D.2 descends to an isomorphism $\overline{\Phi} \colon \overline{\mathcal{F}}_n \to \overline{\mathcal{V}}_{\mathcal{P}}$ such that $\overline{\Phi}(\overline{\mathcal{SM}}_n) = \overline{\mathcal{V}}_{\mathcal{P}}^+$.

For a set function $F \in \overline{\mathcal{F}}_n$, the set of decompositions $\mathcal{D}(F) := \{(G, H) \in \overline{\mathcal{SM}}_n \times \overline{\mathcal{SM}}_n \mid F = G - H\}$ is a polyhedron.

**Corollary D.10.** *A decomposition $(G, H)$ is irreducible if and only if $(G, H)$ is contained in a bounded face of $\mathcal{D}(\mathcal{F})$.*

*Proof.* The extension of $\overline{\Phi}$ to the cartesian product $\overline{\Phi} \times \overline{\Phi} \colon \overline{\mathcal{V}}_{\mathcal{P}} \times \overline{\mathcal{V}}_{\mathcal{P}} \to \overline{\mathcal{F}}_n \times \overline{\mathcal{F}}_n$ is an isomorphism. Then the statement follows from the fact that $\mathcal{D}(F) = (\overline{\Phi} \times \overline{\Phi})(\mathcal{D}_{\mathcal{P}}(\Phi^{-1}(F)))$ and Theorem 3.8. $\qquad \square$

**Definition D.11.** For a submodular function $F \colon 2^{[n]} \to \mathbb{R}$, the *base polytope* $B(F)$ is defined as
$$B(F) := \{x \in \mathbb{R}^n \mid \sum_{i \in S} x_i \leq F(S) \, \forall S \subset [n], \sum_{i \in [n]} x_i = F([n])\}.$$

Since we factored out modular functions, we can assume without loss of generality that a set function $F \in \overline{\mathcal{F}}_n$ is *normalized*, that is, $F(\emptyset) = 0$. In this case, $f = \Phi^{-1}(F)$ is positively homogeneous. For the remaining chapter, we will assume all set functions to be normalized and all CPWL functions to be positively homogeneous. If $F$ is submodular, $f$ agrees with the support function of the base polytope $B(F)$, and therefore $B(F)$ is the Newton polytope $\mathrm{Newt}(f)$ of the Lovász extension $f$ (see e.g. Aguiar & Ardila (2017) Theorem 12.3.). The Newton polytopes of functions that differ by a linear map are a translation of each other and modular functions correspond to linear functions. Hence, if we denote by $\overline{\mathcal{B}}_n$ the set of base polytopes in $\mathbb{R}^n$ modulo translation, the maps $B \colon \overline{\mathcal{F}}_n \to \overline{\mathcal{B}}_n, F \mapsto B(F)$ and $\mathrm{Newt} \colon \overline{\mathcal{V}}_{\mathcal{P}} \to \overline{\mathcal{B}}_n, f \mapsto \mathrm{Newt}(f)$ are well defined and we obtain the following diagramm:

$$
\begin{array}{ccc}
\overline{\mathcal{F}}_n & \xrightarrow{\ B\ } & \overline{\mathcal{B}}_n \\
{\scriptstyle \overline{\Phi}} \downarrow & \nearrow {\scriptstyle \mathrm{Newt}} & \\
\overline{\mathcal{V}}_{\mathcal{P}} & &
\end{array}
$$

In this setting, we call a decomposition $(G, H) \in \mathcal{D}(F)$ *minimal*, if it is not *dominated* by any other decomposition, that is, if there is no other decomposition $(G', H') \in \mathcal{D}(F)$ where $B(G')$ has at most as many vertices as $B(G)$, $B(H')$ has at most as many vertices as $B(H)$, and one of the two has strictly fewer vertices.

For a tuple of submodular functions $(G, H) \in \overline{\mathcal{SM}} \times \overline{\mathcal{SM}}$, let $(\mathcal{P}_G, \mathcal{P}_H)$ be the tuple of the normal fans of the base polytopes $B(G)$ and $B(H)$. A decomposition $(G', H') \in \mathcal{D}(F)$ is called a *(non-trivial) coarsening* of $(G, H) \in \mathcal{D}(F)$ if $\mathcal{P}_{G'}$ and $\mathcal{P}_{H'}$ are coarsenings of $\mathcal{P}_G$ and $\mathcal{P}_H$, respectively (and at least one of them is a non-trivial coarsening).

**Corollary D.12.** $(G, H) \in \mathcal{D}(F)$ *is a vertex if and only if there is no non-trivial coarsening of $(G, H)$.*

*Proof.* Since $g = \Phi^{-1}(G)$ and $h = \Phi^{-1}(H)$ are the support functions of the base polytopes $B(G)$ respectively $B(H)$, the tuple of normal fans $(\mathcal{P}_G, \mathcal{P}_H)$ agrees with the tuple $(\mathcal{P}_g, \mathcal{P}_h)$ of the unique coarsest polyhedral complexes compatible with $g$ and $h$. Hence by Theorem 3.11, there is no non-trivial coarsening of $(G, H)$ if and only if $(g, h)$ is a vertex of $\mathcal{D}_{\mathcal{P}}(f)$ which is the case if and only if $(G, H)$ is a vertex of $\mathcal{D}(F)$. $\qquad \square$

**Corollary D.13.** *For a normalized set function $F \in \overline{\mathcal{F}}_n$, a minimal decomposition of $F$ is a vertex of $\mathcal{D}(F)$.*

*Proof.* If $(G, H)$ is not a vertex, then there is a coarsening $(G', H') \in \mathcal{D}(F)$ of $(G, H)$ implying that $(G', H')$ dominates $(G, H)$. $\qquad \square$

The following example shows that the Lovász extensions of cut functions are hyperplane functions thus admit a unique minimal decomposition into submodular functions, which are themselves cut functions. In particular, the Lovász extensions of the decomposition have at most as many pieces as the Lovász extensions of the original cut function.

**Example D.14** (Minimal decompositions of cut functions). Let $G = (V, E)$ be a graph where $V = [n]$ and $c \colon E \to \mathbb{R}$ a weight function on the edges. Let $F \in \mathcal{F}_n$ be the cut function given by $F(S) = \sum_{\{u,v\} \in \delta(S)} c(\{u, v\})$, where $\delta(S) := \{\{u, v\} \in E \mid u \in S, v \in V \setminus S\}$. The function $f := \Phi^{-1}(F) \in \mathcal{V}_\mathcal{P}$ is given by $f(x) = \sum_{\{u,v\} \in E} c(\{u, v\}) \cdot f_{u,v}(x)$, where $f_{u,v}(x) = \max\{x_u - x_v, x_v - x_u\}$. To see this, first note that $f \in \mathcal{V}_\mathcal{P}$. Thus, it suffices to check that $F(S) = f(\mathbb{1}_S)$ for all $S \subseteq [n]$, which follows due to the observation that

$$f_{u,v}(\mathbb{1}_S) = \begin{cases} 1 & \{u, v\} \in \delta(S) \\ 0 & \{u, v\} \notin \delta(S) \end{cases}$$

Hence, Example A.2 implies that the functions

$$g = \sum_{c(\{u,v\}) > 0} c(\{u, v\}) \cdot f_{u,v} \text{ and } h = \sum_{c(\{u,v\}) < 0} c(\{u, v\}) \cdot f_{u,v}$$

form the unique minimal decomposition of $f$. Thus, $G = \Phi(g)$ and $H = \Phi(h)$, the submodular functions given by

$$G(S) = \sum_{\substack{\{u,v\} \in \delta(S) \\ c(\{u,v\}) > 0}} c(\{u, v\}) \text{ and } H(S) = \sum_{\substack{\{u,v\} \in \delta(S) \\ c(\{u,v\}) < 0}} c(\{u, v\})$$

are the unique minimal decompositions of $F$ into submodular functions.

# E PROOFS

## E.1 PROOF OF LEMMA 3.1

*Proof.* Let $f, g$ be CPWL functions which are compatible with $\mathcal{P}$, and $\lambda, \mu \in \mathbb{R}$. Then for any $P \in \mathcal{P}_n$ holds $(\lambda f + \mu g)|_P = \lambda f|_P + \mu g|_P$, which is an affine function restricted to $P$. Thus, the set $\mathcal{V}_\mathcal{P}$ of CPWL functions compatible with $\mathcal{P}$ forms a linear subspace of the space of continuous functions. $\square$

## E.2 PROOF OF LEMMA 3.2

*Proof.* Let $f \in \mathcal{V}_\mathcal{P}$. Since $\mathcal{P}$ is a complete polyhedral complex, for every $\sigma \in \mathcal{P}^{n-1}$, there are $P, Q \in \mathcal{P}^n$ such that $\sigma = P \cap Q$. Let $a_P, a_Q \in \mathbb{R}^n$ and $b_P, b_Q \in \mathbb{R}$ such that $f|_P(x) = \langle a_P, x \rangle + b_P$ and $f|_Q(x) = \langle a_Q, x \rangle + b_Q$. Consider the linear map $\phi \colon \mathcal{V}_\mathcal{P} \to \mathcal{W}_\mathcal{P}$ given by

$$w_f(\sigma) := \langle e_{P/\sigma}, a_P \rangle + \langle e_{Q/\sigma}, a_Q \rangle = \langle e_{P/\sigma}, a_P - a_Q \rangle.$$

Note that if $f$ is locally convex at $\sigma$, then $\langle e_{P/\sigma}, a_P - a_Q \rangle = \|a_P - a_Q\|_2$ and if $f$ is locally concave at $\sigma$, then $\langle e_{P/\sigma}, a_P - a_Q \rangle = -\|a_P - a_Q\|_2$. The proof proceeds analogously to the case where $f$ has only convex breakpoints and the coefficients of the affine maps are rational. In this case, the lemma follows from the structure theorem of tropical geometry. See Maclagan & Sturmfels (2015) Proposition 3.3.2 for a proof that $w_f \in \mathcal{W}_\mathcal{P}$ and Maclagan & Sturmfels (2015) Proposition 3.3.10 for a proof that $\phi$ is surjective. Here, we present an adjusted proof (to not necessarily convex functions and irrational coefficients). First, we check that $w_f \in \mathcal{W}_\mathcal{P}$. Let $\tau \in \mathcal{P}^{n-2}$ and $\{P_1, \ldots, P_m\} = \mathrm{star}_\mathcal{P}(\tau) \cap \mathcal{P}^n$ and $\{\sigma_1, \ldots, \sigma_m\} = \mathrm{star}_\mathcal{P}(\tau) \cap \mathcal{P}^{n-1}$ be ordered in a cyclic way, that is, $P_i \cap P_{i+1} = \sigma_i$ for $i \in [m]$, where $P_{m+1} = P_1$. Note that, since $f$ is continuous, we have that $a_{P_i} - a_{P_{i+1}} \in \mathrm{span}(e_{P_i/\sigma_i})$. The linear map $T_\tau \colon \mathrm{aff}(\tau)^\perp \to \mathrm{aff}(\tau)^\perp$ satisfying $T_\tau(e_{P_i/\sigma_i}) = e_{\sigma_i/\tau}$ (given by a rotation matrix)

is an automorphism, implying that

$$\sum_{\substack{\sigma \supset \tau \\ \sigma \in \mathcal{P}^{n-1}}} w_f(\sigma) \cdot e_{\sigma/\tau} = \sum_{i=1}^{m} w_f(\sigma_i) \cdot T_\tau(e_{P_i/\sigma_i})$$

$$= T_\tau \left( \sum_{i=1}^{m} \langle e_{P_i/\sigma_i}, a_{P_i} - a_{P_{i+1}} \rangle \cdot e_{P_i/\sigma_i} \right)$$

$$= T_\tau \left( \sum_{i=1}^{m} \langle e_{P_i/\sigma_i}, e_{P_i/\sigma_i} \rangle \cdot (a_{P_i} - a_{P_{i+1}}) \right)$$

$$= T_\tau(0) = 0$$

We proceed by showing that the map $\phi$ is surjective and its kernel is precisely $\mathrm{Aff}(\mathbb{R}^n)$ and therefore, it induces an isomorphism between $\overline{\mathcal{V}}_\mathcal{P}$ and $\mathcal{W}_\mathcal{P}$.

The kernel of $\phi$ is $\mathrm{Aff}(\mathbb{R}^n)$ since $w_f(\sigma) = \langle e_{P/\sigma}, a_P \rangle + \langle e_{Q/\sigma}, a_Q \rangle = \langle e_{P/\sigma}, a_P - a_Q \rangle = 0$ if and only if $a_P - a_Q = 0$ due to the fact that $a_P - a_Q \in \mathrm{span}(e_{P/\sigma})$. Due to the continuity of $f$, this also implies that $b_P = b_Q$ and hence $f|_Q = f|_P$ and therefore the map $f$ is affine linear.

To show surjectivity, let $w \in \mathcal{W}_\mathcal{P}$. We aim to find an $f \in \mathcal{V}_\mathcal{P}$ such that $w = \phi(f)$. Let $\mathcal{P}^n = \{P_1, \dots P_k\}$ and for $P \in \mathcal{P}^n, \sigma \in \mathcal{P}^{n-1}$, let $b_{P/\sigma} \in \mathbb{R}$ such that $\sigma$ is contained in the hyperplane $\{x \in \mathbb{R}^n \mid \langle e_{P/\sigma}, x \rangle + b_{P/\sigma} = 0\}$ and define the function $f_{P/\sigma} \colon \mathbb{R}^n \to \mathbb{R}$ by $f_{P/\sigma}(x) = \langle e_{P/\sigma}, x \rangle + b_{P/\sigma}$. Since $\mathcal{P}$ is complete, the graph $G = (V, E)$ given by $V = \{1, \dots, k\}$ and $E = \{\{i, j\} \mid P_j \cap P_i \in \mathcal{P}^{n-1}\}$ is connected. Start by defining the function $f|_{P_1} = 0$. For $1 < i \leq k$, let $(j_1, \dots j_m)$ be a path from 1 to $i$ and for $\ell \in [m-1]$, let $\sigma_\ell = P_{j_\ell} \cap P_{j_{\ell+1}}$ and define the function

$$f|_{P_i} = \sum_{\ell=1}^{m-1} w(\sigma_\ell) \cdot f_{P_{j_\ell}/\sigma_\ell}.$$

First, we argue that $f|_{P_i}$ is well-defined, that is, the definition of $f|_{P_i}$ does not depend on the path from vertex 1 to $i$. Equivalently, it suffices to show that for any cycle $(i_1, \dots, i_m)$ in $G$ with $i_1 = i_m$, it holds that $\sum_{\ell=1}^{m-1} w(\sigma_\ell) f_{P_{i_\ell}/\sigma_\ell} = 0$, where $\sigma_\ell = P_{i_\ell} \cap P_{i_{\ell+1}}$. Since $\mathcal{P}$ is complete any cycle decomposes into cycles $(i_1, \dots, i_m)$ corresponding to the star of a cone $\tau \in \mathcal{P}^{n-2}$, that is, $\{\sigma_1, \dots, \sigma_{m-1}\} = \{\sigma \in \mathcal{P}^{n-1} \mid \sigma \supset \tau\}$. Since $T_\tau$ is an automorphism, it holds that $\sum_{\ell=1}^{m-1} w(\sigma_\ell) \cdot e_{\sigma_\ell/\tau} = 0$ if and only if $\sum_{\ell=1}^{m-1} w(\sigma_\ell) \cdot e_{P_{i_\ell}/\sigma_\ell} = 0$

So, let $x \in \mathbb{R}^n$ be arbitrary and $x' \in \mathrm{aff}(\tau)$ and $x'' \in \mathrm{aff}(\tau)^\perp$ such that $x = x' + x''$. Since $w$ is in $\mathcal{W}_\mathcal{P}$, it holds that $\sum_{\substack{\sigma \supset \tau \\ \sigma \in \mathcal{P}^{n-1}}} w(\sigma) \cdot e_{\sigma/\tau} = 0$ and hence it follows that

$$\sum_{\ell=1}^{m-1} w(\sigma_\ell) f_{P_{i_\ell}/\sigma_\ell}(x) = \sum_{\ell=1}^{m-1} w(\sigma_\ell) \cdot \langle e_{P_{i_\ell}/\sigma_\ell}, x' + x'' \rangle + b_{P_{i_\ell}/\sigma_\ell}$$

$$= \sum_{\ell=1}^{m-1} w(\sigma_\ell) \cdot \langle e_{P_{i_\ell}/\sigma_\ell}, x'' \rangle$$

$$= \langle \sum_{\ell=1}^{m-1} w(\sigma_\ell) \cdot e_{P_{i_\ell}/\sigma_\ell, x''} \rangle$$

$$= 0$$

By definition, $f$ is a CPWL function and compatible with $\mathcal{P}$ and hence in $\mathcal{V}_\mathcal{P}$. To see that $w = \phi(f)$, let $P, Q \in \mathcal{P}^n$ such that $\sigma = P \cap Q \in \mathcal{P}^{n-1}$. Then it holds that $a_P - a_Q = w(\sigma) \cdot e_{P/\sigma}$ and hence

$$w_f(\sigma) = \langle e_{P/\sigma}, a_P \rangle + \langle e_{Q/\sigma}, a_Q \rangle = \langle e_{P/\sigma}, a_P - a_Q \rangle = \langle e_{P/\sigma}, w(\sigma) \cdot e_{P/\sigma} \rangle = w(\sigma),$$

finishing the proof. $\qquad\square$

### E.3 Corollary E.1

**Corollary E.1.** $\mathcal{V}_{\mathcal{P}}$ *is finite-dimensional.*

*Proof.* By Lemma 3.2, we have that $\overline{\mathcal{V}}_{\mathcal{P}} = \mathcal{V}_{\mathcal{P}}/\operatorname{Aff}(\mathbb{R}^n) \cong \mathcal{W}_{\mathcal{P}}$. Thus, the dimension of $\mathcal{V}_{\mathcal{P}}$ is bounded from above by $\dim(\mathcal{W}_{\mathcal{P}}) + \dim(\operatorname{Aff}(\mathbb{R}^n)) \leq |\mathcal{P}^{n-1}| + (n+1)$. □

### E.4 Proof of Proposition 3.3

*Proof.* The function $f$ is convex if and only if it is locally convex around every $x \in \mathbb{R}^n$. If $x$ is in the relative interior of some $P \in \mathcal{P}^n$, then this is clearly satisfied since the function is locally affine linear. Now, assume that $f$ is not locally convex around a $x \in \tau$ for some $\tau \in \mathcal{P}^{n-2}$. In other words, there are $z, y \in \mathbb{R}^n$ such that $f(\lambda z + (1-\lambda)y) > \lambda f(z) + (1-\lambda)f(y)$ and such that the line between $x$ and $y$ intersects $\tau$. Let $L$ be the Lipschitz constant of $f$ and $\delta := f(\lambda z + (1-\lambda)y) - \lambda f(z) + (1-\lambda)f(y) > 0$. Let $\varepsilon := \frac{\delta}{4L} > 0$. Then there are $v, w \in \mathbb{R}^n$ with $\|v\|, \|w\| < \varepsilon$ such that the line between $z + v$ and $y + w$ does not intersect any face $\tau \in \mathcal{P}^{n-2}$. But then,

$$
\begin{aligned}
f(\lambda(z+v) + (1-\lambda)(y+w)) &\geq f(\lambda z + (1-\lambda)y) - L(\|\lambda v\| + \|(1-\lambda)w\|) \\
&> f(\lambda z + (1-\lambda)y) - 2L\varepsilon \\
&= \delta + \lambda f(z) + (1-\lambda)f(y) - 2L\varepsilon \\
&\geq \delta + \lambda f(z+v) + (1-\lambda)f(y+w) - 2L\varepsilon - 2L\varepsilon \\
&= \lambda f(z+v) + (1-\lambda)f(y+w)
\end{aligned}
$$

and there must be a $x'$ in the relative interior of some $\sigma \in \mathcal{P}^{n-1}$ such that $f$ is not locally convex around $x'$. Hence, $f$ is convex if and only $f$ is locally convex around every $\sigma \in \mathcal{P}^{n-1}$, that is, $f$ is locally convex around every $x$ in the relative interior of $\sigma$. For any such $x$, there is a $\lambda > 0$ such that $x + \lambda \cdot e_{P/\sigma} \in P$ and $x + \lambda \cdot e_{Q/\sigma} \in Q$, by construction of $e_{P/\sigma}$ and $e_{Q/\sigma}$. Recall from the proof of Lemma 3.2 that $w_f(\sigma) = \langle e_{P/\sigma}, a_P \rangle + \langle e_{Q/\sigma}, a_Q \rangle$, where $f|_P(x) = \langle a_P, x \rangle + b_P$ and $f|_Q(x) = \langle a_Q, x \rangle + b_Q$. Since $P, Q \in \mathcal{P}^n$ and $\|e_{P/\sigma}\| = \|e_{Q/\sigma}\| = 1$, we have that $x$ is the midpoint of $x + \lambda \cdot e_{P/\sigma}$ and $x + \lambda \cdot e_{Q/\sigma}$. Therefore, $f$ is convex if and only if $f(x) \leq \frac{1}{2}f(x + \lambda \cdot e_{P/\sigma}) + \frac{1}{2}f(x + \lambda \cdot e_{Q/\sigma})$. Equivalently,

$$
0 \leq f(x + \lambda \cdot e_{P/\sigma}) + f(x + \lambda \cdot e_{Q/\sigma}) - 2f(x) = \lambda(\langle e_{P/\sigma}, a_P \rangle + \langle e_{Q/\sigma}, a_Q \rangle) = \lambda \cdot w_f(\sigma).
$$

If $\mathcal{P} = \mathcal{P}_f$, then we have strict local convexity at every $\sigma \in \mathcal{P}^{n-1}$, which means a strict inequality in the inequality above. □

### E.5 Lemma E.2

**Lemma E.2.** $\overline{\mathcal{V}}_{\mathcal{P}}^{+}$ *forms a polyhedral cone in* $\overline{\mathcal{V}}_{\mathcal{P}}$.

*Proof.* Lemma 3.2 and Proposition 3.3 imply that the set of convex functions in $\overline{\mathcal{V}}_{\mathcal{P}}$ satisfies $\overline{\mathcal{V}}_{\mathcal{P}}^{+} \cong \mathcal{W}_{\mathcal{P}}^{+} := \bigcap_{\sigma \in \mathcal{P}^{n-1}} \{w \in \mathcal{W}_{\mathcal{P}} \mid w(\sigma) \geq 0\}$. This is a finite intersection of linear inequalities, so $\mathcal{W}_{\mathcal{P}}^{+}$ is a polyhedral cone. Moreover, "$\cong$" is a linear isomorphism, which implies that $\overline{\mathcal{V}}_{\mathcal{P}}^{+}$ is a polyhedral cone. □

### E.6 Lemma E.3

**Lemma E.3.** *Let $\mathcal{P}$ be a regular polyhedral complex. Then every CPWL function compatible with $\mathcal{P}$ can be written as a difference of two convex CPWL functions that are also compatible with $\mathcal{P}$. In particular, $\overline{\mathcal{V}}_{\mathcal{P}} = \operatorname{span}(\overline{\mathcal{V}}_{\mathcal{P}}^{+})$.*

*Proof.* Let $f \in \mathcal{V}_{\mathcal{P}}$ be an arbitrary function. Since $\mathcal{P}$ is regular, by definition there exists a convex function $g \in \mathcal{V}_{\mathcal{P}}$ such that $\mathcal{P} = \mathcal{P}_g$. Proposition 3.3 implies that $w_g(\sigma) > 0$ for all

$\sigma \in \mathcal{P}^{n-1}$. For sufficiently large $\lambda > 0$, it follows that $w_{f+\lambda g} \geq 0$ and thus $f = (f + \lambda g) - \lambda g$ is a representation of $f$ as a difference of two compatible, convex functions, as desired. $\quad\square$

### E.7 PROOF THEOREM 3.5

*Proof.* For the set of decompositions holds

$$\mathcal{D}_{\mathcal{P}}(f) = \{(g, h) \mid g \in \overline{\mathcal{V}}_{\mathcal{P}}^+, h \in \overline{\mathcal{V}}_{\mathcal{P}}^+, f = g - h\} = (\overline{\mathcal{V}}_{\mathcal{P}}^+ \times \overline{\mathcal{V}}_{\mathcal{P}}^+) \cap H_f.$$

For the projection we have

$$\pi(\mathcal{D}_{\mathcal{P}}(f)) = \pi(\{(g, g - f) \mid g \in \overline{\mathcal{V}}_{\mathcal{P}}^+, g - f \in \overline{\mathcal{V}}_{\mathcal{P}}^+\}) = \{g \mid g \in \overline{\mathcal{V}}_{\mathcal{P}}^+, g \in f + \overline{\mathcal{V}}_{\mathcal{P}}^+\} = \overline{\mathcal{V}}_{\mathcal{P}}^+ \cap (f + \overline{\mathcal{V}}_{\mathcal{P}}^+).$$

$$\square$$

### E.8 PROOF OF THEOREM 3.8

The statement follows from a more general statement about polyhedra. Recall that any polyhedron $P$ can written as the Minkowski sum

$$P = Q + C = \{q + c \mid q \in Q, c \in C\}$$

where $Q$ is a bounded polytope, and $C$ a unique polyhedral cone, the *recession cone* of $P$.

**Proposition E.4.** *A point $x \in P$ is contained in a bounded face of $P$ if and only if $x - c \notin P \ \forall c \in C \setminus \{0\}$.*

*Proof.* Any face of the polyhedron $P$ is of the form

$$P^u = \{x \in P \mid \langle x, u \rangle \geq \langle y, u \rangle \ \forall y \in P\},$$

and for Minkowski sums holds $P^u = Q^u + C^u$. Let $x \in P$ be a point contained in a bounded face $P^u$ of $P$. Since $P^u$ is bounded, we have that $P^u = Q^u + C^u$ with $C^u = \{0\}$ being the unique bounded face of $C$. Thus, $\langle c, u \rangle < \langle 0, u \rangle$ for all $c \in C \setminus \{0\}$. This implies that $\langle x - c, u \rangle = \langle x, u \rangle - \langle c, u \rangle > \langle x, u \rangle$ and therefore, by definition of $P^u$, we have that $x - c \notin P$.

Conversely, suppose that $x \in P$ is not contained in a bounded face. We want to show that there exists some direction $c \in C \setminus \{0\}$ such that $x - c \in P$. Since $x$ is not contained in a bounded face, it is contained in the relative interior of an unbounded face $F$ (where possibly $F = P$). Since the face is unbounded, it contains a ray $x + \mathbb{R}c$ for some direction $c \in C$. On the other hand, since $x \in \text{int}(F)$, we have that $x - \varepsilon c \in F$ for $\varepsilon > 0$ small enough. As $C$ is a cone, we have that $\varepsilon c \in C$, which finishes the proof. $\quad\square$

*Proof of Theorem 3.8.* Since $\pi$ induces a bijection between $\mathcal{D}_{\mathcal{P}}(f)$ and its image, this is also a bijection between bounded faces. By Theorem 3.5, $\pi(\mathcal{D}_{\mathcal{P}}(f))$ is a polyhedron with recession cone $\overline{\mathcal{V}}_{\mathcal{P}}^+$. Proposition E.4 implies that $g$ is contained in a bounded face if and only if there exists no convex function $\phi \in \overline{\mathcal{V}}_{\mathcal{P}}^+$ such that $g - \phi \in \pi(\mathcal{D}_{\mathcal{P}}(f))$. Therefore, $\pi^{-1}(g) = (g, h), h = g - f$ is contained in a bounded face of $\mathcal{D}_{\mathcal{P}}(f)$ if and only if there is no $\phi \in \overline{\mathcal{V}}_{\mathcal{P}}^+$ such that $(g - \phi, g - f - \phi) = (g - \phi, h - \phi) \notin \mathcal{D}_{\mathcal{P}}(f)$. Since $(g - \phi) - (h - \phi) = f$, this is equivalent to $g - \phi$ or $h - \phi$ being nonconvex, i.e., $(g, h)$ is reduced. $\quad\square$

### E.9 PROOF OF LEMMA 3.10

*Proof.* First note that $B(g) := \bigcup\limits_{\sigma \in \text{supp}_{\mathcal{P}}(g)} \sigma$ are exactly the points where $g$ is not affine linear. Hence, the closures of the connected components of the complement of $B(g)$ are the maximal polyhedra of the unique coarsest polyhedral complex $\mathcal{P}_g$ compatible with $g$.

Let $\text{supp}_{\mathcal{P}}(g') \subseteq \text{supp}_{\mathcal{P}}(g)$. Equivalently, for the complement holds $(\mathbb{R}^n \setminus B(g)) \subseteq (\mathbb{R}^n \setminus B(g'))$, and the same holds for the closures of the (open) connected components, i.e., the maximal faces in $\mathcal{P}_g^n$ and $\mathcal{P}_{g'}^n$. In other words, this is equivalent to that for every face $P \in \mathcal{P}_g^n$

there exists some $P' \in \mathcal{P}_{g'}^n$ such that $P \subseteq P'$. Thus, $\mathrm{supp}_{\mathcal{P}}(g') \subseteq \mathrm{supp}_{\mathcal{P}}(g)$ if and only if $\mathcal{P}_{g'}$ is a coarsening of $\mathcal{P}_g$.

The coarsening is non-trivial if and only if there is a $P' \in \mathcal{P}_{g'}^n$ such that there is no $P \in \mathcal{P}_g^n$ with $P' \subseteq P$. This is the case if and only if there is a $\sigma \in \mathcal{P}_g^{n-1}$ that intersects the interior of $P'$, which occurs if and only if $\sigma \in \mathrm{supp}_{\mathcal{P}}(g) \setminus \mathrm{supp}_{\mathcal{P}}(g')$. $\qquad\square$

### E.10    Proof of Theorem 3.11

We first prove the following proposition that relates coarsenings of the decompositions to inclusion relations of the minimal faces that contain the decompositions.

**Proposition E.5.** *For $(g, h) \in \mathcal{D}_{\mathcal{P}}(f)$, let $F$ be the minimal face of $\mathcal{D}_{\mathcal{P}}(f)$ containing $(g, h)$. Then $(g', h')$ is a coarsening of $(g, h)$ if and only if there is a face $G$ of $\mathcal{D}_{\mathcal{P}}(f)$ with $G \subseteq F$ such that $(g', h') \in G$. The coarsening is non-trivial if and only if $F \neq G$.*

*Proof.* For a face $F$, let $\mathcal{G}_F = \{\sigma \in \mathcal{P}^{n-1} \mid w_g(\sigma) = 0 \text{ for all } (w_g, w_h) \in F\}$ and $\mathcal{H}_F = \{\sigma \in \mathcal{P}_f^{n-1} \mid w_h(\sigma) = 0 \text{ for all } (w_g, w_h) \in F\}$ be the set of facets where the corresponding inequalities ensuring convexity of the functions $g$ and $h$ are tight. It is not hard to see that $G \subseteq F$ if and only if $\mathcal{G}_F \subseteq \mathcal{G}_G$ and $\mathcal{H}_F \subseteq \mathcal{H}_G$. In other words, if $(g', h')$ is contained in a face $G \subseteq F$, then one can move from $(g, h)$ to $(g', h')$ without losing tight inequalities. Hence, Lemma 3.10 implies that $(g', h')$ is a coarsening of $(g, h)$. If $G \subset F$, then either $\mathcal{G}_F \subset \mathcal{G}_G$ or $\mathcal{H}_F \subset \mathcal{H}_G$. Thus, another inequality becomes tight when moving from $(g, h)$ to $(g', h')$ implying that the coarsening is non-trivial.

For the converse direction, let $(g', h')$ be a coarsening of $(g, h)$, which in particular means that $g'$ and $h'$ are compatible with $\mathcal{P}$. Hence, $f = g' - h'$ implies that $(g', h') \in \mathcal{D}_{\mathcal{P}}(f)$. Now, assume that there is no face $G \subseteq F$ such that $(g', h') \in G$. Then the line between $(g, h)$ and $(g', h')$ is not contained in $F$. Thus, a tight inequality gets lost when moving from $(g, h)$ towards $(g', h')$. Hence, without loss of generality, there is a $\sigma \in \mathrm{supp}_{\mathcal{P}}(g') \setminus \mathrm{supp}_{\mathcal{P}}(g)$, which according to Lemma 3.10 is a contradiction to $(g', h')$ being a coarsening of $(g, h)$. $\quad\square$

*Proof of Theorem 3.11.* 1 and 2 are equivalent by Proposition E.5. 3 trivially implies 2. Hence, it remains to show 1 $\implies$ 3. Assume that there is a polyhedral complex $\mathcal{Q}$ compatible with $g$ and $h$ such that $(g, h)$ is not a vertex of $\mathcal{D}_{\mathcal{Q}}(f)$. Then there is vertex $(g', h')$ of $\mathcal{D}_{\mathcal{Q}}(f)$ contained in the face containing $(g, h)$. By Proposition E.5, it follows that $(g', h')$ is a non-trivial coarsening of $(g, h)$. $\qquad\square$

### E.11    Proof of Theorem 3.13

*Proof.* If $(g, h)$ is not not a vertex, the by Theorem 3.11, there is a coarsening $(g', h')$ of $(g, h)$. Thus, $(g, h)$ is dominated by $(g, h)$ and therefore not minimal. $\qquad\square$

### E.12    Proof of Proposition 3.14

*Proof.* As $\mathcal{D}_{\mathcal{P}}(f)$ is nonempty, there must exist a minimal decomposition. By Theorem 3.13, every minimal decomposition must be a vertex. As there is only one vertex, it must coincide with the unique minimal decomposition. $\qquad\square$

### E.13    Lemma E.6

**Lemma E.6.** *Let $C \subset \mathbb{R}^d$ be a convex, pointed polyhedral cone. If $C$ is simplicial then $C \cap (C + t)$ is a (potentially shifted) cone, i.e. a polyhedron with a single vertex, for any translation $t$. If $C$ is not simplicial, then $C \cap (C + t)$ is a (shifted) cone if $t \in C$.*

*Proof.* If $C$ is a simplicial full-dimensional cone, then it is the image of the nonnegative orthant under an affine isomorphism. Thus, it suffices to show that $C \cap (C + t)$ is a shifted cone for $C = \mathbb{R}_{\geq 0}^d$. Let $\hat{t} \in \mathbb{R}^d$ such that $\hat{t}_i = \max(t_i, 0)$. Then

$$C \cap (C + t) = \{x \mid x_i \geq 0 \text{ and } x_i \geq t_i\} = \{x \mid x_i \geq \hat{t}_i\} = C + \hat{t}.$$

On the other hand, if $C$ is an arbitrary polyhedral cone and $t \in C$, then $C \cap (C+t) = C+t$, and hence a shifted cone. $\square$

**Example E.7.** The converse of the second statement from Lemma E.6 does not hold, that is, $t \notin C$ does not imply that $C \cap (C+t)$ is not a shfited cone. Indeed, let $C = \text{cone}\left( \begin{pmatrix} 1 \\ 0 \\ 0 \end{pmatrix}, \begin{pmatrix} 1 \\ 1 \\ 0 \end{pmatrix}, \begin{pmatrix} 1 \\ 0 \\ 1 \end{pmatrix}, \begin{pmatrix} 1 \\ 1 \\ 1 \end{pmatrix} \right)$ and $t = \begin{pmatrix} 0 \\ 1 \\ 1 \end{pmatrix}$. Then $C \cap (C+t) = C + \begin{pmatrix} 1 \\ 1 \\ 1 \end{pmatrix}$ is a shifted cone. On the other hand, the choice $t = \begin{pmatrix} 0 \\ 1 \\ 2 \end{pmatrix}$ yields the unbounded polyhedron $C \cap (C+t) = \text{conv}\left( \begin{pmatrix} 2 \\ 2 \\ 2 \end{pmatrix}, \begin{pmatrix} 2 \\ 1 \\ 2 \end{pmatrix} \right) + C$, which has two vertices and one line segment as bounded faces.

### E.14 Proof of Proposition 3.15

*Proof.* Let $\mathcal{Q}$ be any regular complete complex that is compatible with $f$. Then, $g$ and $h$ are as well compatible with $\mathcal{Q}$, since $\text{supp}_{\mathcal{P}}(g), \text{supp}_{\mathcal{P}}(h) \subseteq \text{supp}_{\mathcal{P}}(f)$ implies that $\text{supp}_{\mathcal{Q}}(g), \text{supp}_{\mathcal{Q}}(h) \subseteq \text{supp}_{\mathcal{Q}}(f)$. Let $(g', h') \in \mathcal{D}_{\mathcal{Q}}(f)$. Then it holds that $\text{supp}_{\mathcal{Q}}^{+}(f) \subseteq \text{supp}_{\mathcal{Q}}(g')$ since $w_g - w_f = w_h \geq 0$. Hence, $\text{supp}_{\mathcal{Q}}(g) \subseteq \text{supp}_{\mathcal{Q}}(g')$ and Lemma 3.10 implies that $g$ is a coarsening of $g'$ and analogously it follows that $h$ is a coarsening of $h'$. Therefore, $(g, h)$ is a coarsening of every decomposition and thus by Theorem 3.11 the only vertex of $\mathcal{D}_{\mathcal{Q}}(f)$. Clearly, $g$ and $h$ cannot have more pieces than $f$. $\square$

### E.15 Proof of Theorem 3.18

Before proving this statement, we give a description of the dual cone $(\overline{\mathcal{V}}_{\mathcal{P}}^{+})^{\vee}$. Recall that $\overline{\mathcal{V}}_{\mathcal{P}}^{+} \cong \bigcap_{\sigma \in \mathcal{P}^{n-1}} \{w \in \mathcal{W}_{\mathcal{P}} \mid w(\sigma) \geq 0\}$, i.e., the intersection of the nonnegative orthant $\{w(\sigma) \geq 0\}$ with the linear space $\mathcal{W}_{\mathcal{P}}$. By duality of intersections and sums, it follows that $(\overline{\mathcal{V}}_{\mathcal{P}}^{+})^{\vee}$ is isomorphic to the Minkowksi sum of the nonnegative orthant with $\mathcal{W}_{\mathcal{P}}^{\perp}$. In particular, any $w$ with positive weights $w(\sigma) > 0$ lies in the interior. Theorem 3.18 follows from a general fact about face of polyhedra.

**Lemma E.8.** *Let $C$ be a convex, pointed polyhedral cone and $P$ a polyhedron with recession cone $C$. Then $u \in \text{int}(C^{\vee})$ is a direction in the interior of the dual cone of $C$ if and only if the face $P^u$ of $P$ which is minimized by $u$ is a bounded face.*

*Proof.* Let $P = C + Q$, where $Q$ is a bounded polyhedron. Then for any direction $u$ holds $P^u = C^u + Q^u$. As $C$ is a pointed cone, we have that $C^u$ is bounded if and only if $u \in \text{int}(C^{\vee})$. Since $Q^u$ is bounded for any direction, it follows that $P^u$ is bounded if and only if $u \in \text{int}(C^{\vee})$. $\square$

*Proof of Theorem 3.18.* Recall that every polyhedron $P$ is the set of feasible solutions to some linear program, and that, given a linear functional $u$ such that $P^u$ is bounded, the face $P^u$ coincides with the set of optimal solutions of the linear program. Now, $P = \overline{\mathcal{V}}_{\mathcal{P}}^{+} \cap (\overline{\mathcal{V}}_{\mathcal{P}}^{+} + f)$ is a polyhedron with recession cone $\overline{\mathcal{V}}_{\mathcal{P}}^{+}$. Applying Lemma E.8 yields that for any $u \in (\text{int}((\overline{\mathcal{V}}_{\mathcal{P}}^{+})^{\vee})$, every minimizer in $\pi(\mathcal{D}_{\mathcal{P}}(f))$ lies in a bounded face, which, by Theorem 3.8, are precisely the reduced decompositions. Moreover, if $\pi(\mathcal{D}_{\mathcal{P}}(f))$ contains a unique vertex then by Proposition 3.14 this coincides with the unique minimal decompoition. $\square$

### E.16 Proof of Theorem 6.1

*Proof.* In the convex case, this is literally proven by Hertrich et al. (2021). While Hertrich et al. (2021) have a slightly weaker bound for the nonconvex case, it follows from Koutschan et al. (2023, Thm. 2.4) that the stronger bound for the convex case also applies to the nonconvex case. $\square$

### E.17 Proof of Theorem 6.3

*Proof.* Recall that a convex CPWL function can be written as the maximum of its affine components, that is, $f(x) = \max_{i \in [k]} a_i^T x + b_i$. The idea is to split the $k$ affine components of

$f$ into $r$ groups of size at most $s$, apply Theorem 6.1 to compute the maximum within each group, and then simply compute the maximum of the $r$ group maxima in a straight-forward way.

Let us first focus on computing the maximum of at most $s$ affine components within each of the $r$ groups. By Theorem 6.1, one can achieve this with a neural network of depth $\lceil \log_2(n+1) \rceil + 1$ and overall size $\mathcal{O}(s^{n+1})$. We put all these $r$ neural networks in parallel to each other and add, at the end, the simple neural network computing the maximum of these $r$ maxima according to Arora et al. (2018), which has depth $\lceil \log_2 r \rceil + 1$ and overall size $\mathcal{O}(r)$. Altogether, the resulting neural network will have the desired depth and size. $\quad\square$

### E.18    Proof of Corollary 6.4

*Proof.* By Lemma E.3, $f$ can be decomposed into a difference of two convex functions which are compatible with $\mathcal{P}$. Consequently, each of them has at most $\tilde{q}$ affine components. Applying Theorem 6.3 to both functions separately and simply putting the two corresponding neural networks in parallel, subtracting the outputs, yields a neural network representing $f$ with the desired size bounds. $\quad\square$

