# OpenReview forum: "Decomposition Polyhedra of Piecewise Linear Functions"
_ICLR.cc/2025/Conference — ICLR 2025 Spotlight_

### Official Review · Reviewer_So2m · 2024-10-29

**Soundness:** 4
**Presentation:** 4
**Contribution:** 3
**Rating:** 8
**Confidence:** 3

**Summary:**

The paper studies decomposition of continuous piecewise linear (CWPL) functions into difference of two convex CWPL functions with as few linear pieces as possible.
The contributions of the paper include:
i. A proof that the minimal solution must be vertices of a certain polyhedron which is a set of decompositions. This polyhedron arises as an intersection of two translated cones.
ii. A construction for a unique minimal decomposition for certain CWPL functions in dimension 2 by Tran & Wang (2024) does not extend to higher dimensions.
iii. Applications of the decomposition to submodular function optimization and neural network construction.

**Strengths:**

Paper is well-written. The problem studied is challenging.

**Weaknesses:**

The obvious limitation is that the underlying polyhedral complex is fixed.
There are no bounds shown for the number of pieces in the minimal decomposition. Would it be related to the notion of monomial complexity ? A discussion would be interesting.

**Questions:**

N/A

---

> ### Author Response · Authors · 2024-11-20
>
> We thank the reviewer for the careful examination of our manuscript and their thoughtful comments. We hope to have addressed all items raised in the revision and the responses below. If there are any further questions, please feel free to post them and we are happy to clarify them.
>
> *The obvious limitation is that the underlying polyhedral complex is fixed. There are no bounds shown for the number of pieces in the minimal decomposition. Would it be related to the notion of monomial complexity ? A discussion would be interesting.*
>
> Indeed, as mentioned after Definition 3.12, the notion of minimality coincides with the notion of minimality with respect to monomial complexity as studied in Tran & Wang (2024). Tran & Wang give lower bounds on the monomial complexity in terms of factorization complexity, however, no good bounds are known for either of these notions in general dimensions.

---

### Official Review · Reviewer_UiL3 · 2024-11-02

**Soundness:** 3
**Presentation:** 3
**Contribution:** 3
**Rating:** 8
**Confidence:** 3

**Summary:**

The paper studies the problem of decomposing a given continuous piecewise linear function into the difference of two convex piecewise linear functions. Especially, the authors investigated the problem of finding such a decomposition with the least linear pieces. To tackle this,  a few new theoretical results were proposed and proved. After fixing the polyhedral complex, they study the geometrical properties of such decompositions. Specifically, they show that a minimal solution must be a vertex and hence can be obtained by a simple enumeration. The results can be applied to relu network with 1 hidden layer, statistics functions, submodular functions, and the construction of neural networks.

**Strengths:**

1. The paper presents an innovative approach by linking decomposition problems with polyhedral geometry, leading to the concept of decomposition polyhedra. This is a very novel idea and may inspire many interesting future works.

2. The theoretical analysis of the paper is very solid, providing us with a deep understanding of CPWL decomposition problem.

3. The paper is well-written and well-organized, clearly stating the main contributions and their applications.

4. The applications to ReLU activation, submodular optimization, neural network design, are very interesting.

**Weaknesses:**

The main weakness of the paper is that as stated in Limitations section of the paper. The paper mainly focuses on the development of theories, but does not provide practical implementations and applications of their results.

**Questions:**

1. Definition 3.12. The definition for the minimal decomposition seems same as the Pareto optimality in multi-objective optimization.
2. Hyperplane functions introduced in Def 3.16 satisfy the assumptions. These are functions generated by a ReLU network with 1 hidden layer. Just wondering, does the result also hold for general ReLU networks with more layers?
3. In theorem 3.18, what is the exact meaning of **minimizer**? It is also better to include an explicit linear programming problem described in Theorem 3.18.

---

> ### Author Response · Authors · 2024-11-20
>
> We thank the reviewer for the careful examination of our manuscript and their thoughtful comments. We hope to have addressed all items raised in the revision and the responses below. If there are any further questions, please feel free to post them and we are happy to clarify them.
>
> *The main weakness of the paper is that as stated in Limitations section of the paper. The paper mainly focuses on the development of theories, but does not provide practical implementations and applications of their results.*
>
> While this is indeed a limitation as discussed, we would not necessarily consider it a weakness of the paper. In our opinion, pure theory papers like ours are a valuable contribution to ICLR.
>
> *Definition 3.12. The definition for the minimal decomposition seems same as the Pareto optimality in multi-objective optimization.*
>
> This is correct. We added a remark after the definition.
>
> *Hyperplane functions introduced in Def 3.16 satisfy the assumptions. These are functions generated by a ReLU network with 1 hidden layer. Just wondering, does the result also hold for general ReLU networks with more layers?*
>
> Functions defined via deeper ReLU networks do not necessarily satisfy the assumption of Proposition 3.14. For example, the function $f(x) = \max (0, \min (x_1,x_2))$ is computable by a ReLU neural network with two hidden layers, but not with one hidden layer. The hyperplane extension (Construction B.1) yields a regular polyhedral complex that is compatible with $f$. Tran & Wang (2024) provide two different decompositions of $f$ for which one can verify that they are both vertices of the decomposition polyhedron of the hyperplane extension.
> Note, however, that all the results before Prop. 3.14 are valid for functions defined via deeper networks, too, as they are valid for any CPWL function.
>
> *In theorem 3.18, what is the exact meaning of minimizer? It is also better to include an explicit linear programming problem described in Theorem 3.18.*
>
> A minizer is meant to be an optimal solution in the framework of linear programming.
> We reformulated Theorem 3.18 to make it fit into this framework.

---

> > ### Comment · Reviewer_UiL3 · 2024-11-26
> >
> > I would like to thank the authors for responding to my comments and the revision! It is a very nice pure theory paper and I will increase the score to 8.

---

### Official Review · Reviewer_BUg5 · 2024-11-06

**Soundness:** 3
**Presentation:** 3
**Contribution:** 3
**Rating:** 8
**Confidence:** 3

**Summary:**

This paper studies the fundamental problem of decomposing a continuous piecewise linear (CPWL) function into two convex CPWL functions. This is a rather challenging problem with a long history and important practical applications. The authors adopt a novel perspective to tackle this problem: investigating the space of admissible convex CPWL functions while fixing the underlying pieces. The main contribution is a series of structural results concerning the geometric properties of the so-called (and new) decomposition polyhedra, which are connected to the space of admissible convex CPWL functions. With these structural results, the authors demonstrate some implications for submodular functions and for constructing neural networks according to a given convex CPWL function. Moreover, they refute a recent conjecture on algorithms for computing CPWL functions in the literature.

**Strengths:**

The DC representation of a general CPWL function is an old but fundamental problem with many applications in various engineering fields. This paper proposes an interesting perspective on how to understand and compute the DC components from a given CPWL function. Although I did not have time to check all the proofs in detail, the paper is generally well written, and the results are interesting. In particular, I appreciate the idea of fixing the underlying pieces and the clean characterization of the DC components.

**Weaknesses:**

While the pieces are assumed to be fixed in advance (which is, of course, a limitation, as also explicitly noted by the authors), I believe this work has great potential to motivate further investigation into both the theoretical and algorithmic aspects of the decomposition problem.

My comments are as follows:
* L200, in the definition of $\mathcal{P}_f^n$, I don't think the set {$x:g_i(x)=\max_j g_j(x)$} must be full dimensional. This may depend on the representation of $f$ given in L199. Also, $k=q$ may not be true, as claimed in L202.
* As for regular polyhedra complex, I understand the usage of the existence of convex CPWL in the proof of theorems. I'm curious about the irregular case and it will be very helpful to provide some details or examples to illustrate the existence of irregular polyhedra complex.
* Some notation are used before defined in the whole paper. For example, in Proposition 3.3, it seems the function $w_f$ is not defined until the proof of Proposition 3.2.
* L184, in the definition of CPWL functions, I think you need to require $f$ to be continuous.
* L1283, what is $\phi$ defined here?
* In the proof of Proposition 3.3, I suggest providing more details to justify the equivalence claimed in line 1360. Intuitively, this is correct, but in convex geometry, counterintuitive phenomena can occur, so a rigorous and formal argument would be desirable.
* L752, the function $h$ may not be convex as claimed.
* L180, add a period.
* L146, "wit" should be "with".

**Questions:**

See above.

---

> ### Author Response · Authors · 2024-11-20
>
> We thank the reviewer for the careful examination of our manuscript and their thoughtful comments. We hope to have addressed all items raised in the revision and the responses below. If there are any further questions, please feel free to post them and we are happy to clarify them.
>
> *L200, in the definition of $\mathcal{P}_f^n$, I don't think the set {$x:g_i(x)=\max_j g_j(x)$} must be full dimensional. This may depend on the representation of $f$ given in L199. Also, $k=q$ may not be true, as claimed in L202.*
>
> We changed the sentence to: If f is a convex CPWL function, then it can be uniquely written as the maximum of finitely many affine functions $f(x)=\max_{i\in[k]}g_i(x)$ such that $k=q$
>
>
> *As for regular polyhedra complex, I understand the usage of the existence of convex CPWL in the proof of theorems. I'm curious about the irregular case and it will be very helpful to provide some details or examples to illustrate the existence of irregular polyhedra complex.*
>
> An example of a complete polyhedral complex which is not regular is given by the following variant of an example that is known as the "mother of all counterexamples" in polyhedral geometry. The classical example is depicted in Figure 16.3.2(b) in https://www.csun.edu/~ctoth/Handbook/chap16.pdf. By adding 3 additional unbounded 1-dimensional faces, we obtain a complete polyhedral complex $\mathcal P$:
> - from the bottom left vertex, add a ray towards $(-\infty,-\infty)$
> - from the bottom right vertex, add a ray towards $(\infty, -\infty)$
> - from the top vertex, add a ray towards $(0,\infty)$
> It is not difficult to show that the obtained complex is indeed not regular: If there was a function $f$ with $\mathcal P = \mathcal P_f$, then the existence of the edges connecting the outer and inner triangles imply a chain of inequalities
> $f(v_1)<f(v_2)<f(v_3)<f(v_1)$, where $v_1,v_2,v_3$ are the vertices of the inner triangle.
> Therefore, there exists no convex CPWL function having this polyhedral complex as underlying polyhedral complex, and is thus not within the scope of our article.
>
>
> *Some notation are used before defined in the whole paper. For example, in Proposition 3.3, it seems the function $w_f$ is not defined until the proof of Proposition 3.2.*
>
> Added explanation.
>
>
> *L184, in the definition of CPWL functions, I think you need to require $f$ to be continuous.*
>
> done, added continuous.
>
>
> *L1283, what is $\phi$ defined here?*
>
> $\Phi$ is the linear map defined by the equation in the following line. The remainder of the proof shows that $\Phi$ defined in this way is an isomorphism.
>
>
> *In the proof of Proposition 3.3, I suggest providing more details to justify the equivalence claimed in line 1360. Intuitively, this is correct, but in convex geometry, counterintuitive phenomena can occur, so a rigorous and formal argument would be desirable.*
>
> We added a formal argument why convexity is equivalent to local convexity around the interior of facets.
>
>
>
> *L752, the function $h$ may not be convex as claimed.*
>
> Yes, there was a minus sign missing. We changed it to $h(x) = -\sum_{ \lambda_i < 0} \lambda_i \cdot \max\{\langle x,a_i \rangle+b_i,\langle x,c_i \rangle+d_i\}$.
>
> *L180, add a period.*
>
> done.
>
>
> *L146, "wit" should be "with".*
>
> done.

---

### Official Review · Reviewer_HSgx · 2024-11-09

**Soundness:** 3
**Presentation:** 2
**Contribution:** 2
**Rating:** 5
**Confidence:** 3

**Summary:**

This paper deals with the following problem: given a continuous piecewise linear (CPWL) function, decompose it as the difference of two convex CPWL functions. This line of research is motivated by the DC paradigm for nonconvex optimization.  In this work the authors consider decompositions that are compatible with a given polyhedral complex and are “minimal” (for the definition of minimality, see Definition 3.12). The authors show that the set of all decompositions forms a polyhedron (Theorem 3.5) and that minimal decompositions are at a vertex of this polyhedron(Theorem 3.13). Thus, a minimal decomposition can be found by enumerating the vertices. They also identify a few special cases where there is a unique vertex, so there is no need for enumeration. In terms of applications, in Section 5 they apply their results to submodular functions. In Section 6, they revisit the problem of representing a CPWL function by a ReLU neural network. Their main result there is Corollary 6.4, providing “a smooth tradeoff between size and depth”, for the special class of CPWL functions that are compatible with a regular polyhedral complex.

**Strengths:**

The problem of decomposing CPWL functions as the difference of convex CPWL functions is interesting. No finite procedure is currently known for finding a minimal decomposition. This work provides a new perspective, based on polyhedral geometry, that guarantees finite convergence, but in the special case where the factors have a fixed supporting polyhedral decomposition

**Weaknesses:**

The assumption that the DC decomposition should be with respect to a fixed polyhedral complex seems restrictive and perhaps unmotivated

I feel that the paper is hard to read for non-experts in the area (e.g. many technical definitions with not many accompanying figures to help the reader-there are few, but mostly in the Appendix).

The main result in Section 6 (Corollary 6.4) for representing CPWL functions as NNs, is only applicable to CPWL functions that are compatible with a regular polyhedral complex. It is not clear whether this assumption is (1) natural and (2) easy to satisfy.  The existing decomposition results are applicable to any CPWL function.

**Questions:**

Page 1, Introduction: You write, “CWPL functions play a crucial role in ML.” If I understand correctly, this is the case because they are used as a test case for the universality theorems for NNs, where they can be concretely instantiated in terms of width, depth, etc. Is that correct, or are there more important applications? You do mention submodularity etc in the introduction, maybe elaborate on that somewhat?

page 1: Are there problems where fixing the supporting polyhedral partition in DC decompositions is a natural constraint?

Page 2, Section 1.1: Please point to a specific Theorem as your “main technical result.” I guess that is Theorem 3.13?

Page 2, line 73: “cannot be easily simplified.” Explain why this is important. Also, when you say a “minimal” solution, what are you referring to?
Page 2, line 75: “simply enumerating” seems to indicate there are not that many of them. Please elaborate on this.

page 3. Please give a bivariate example of a CPWL function with two different compatible polyhedral complexes with a different number of maximal facets.

Page 4: Please provide a simple example (with a figure) of a polyhedral complex illustrating (some of) the definitions in Section 2. In particular, at least one example of a polyhedral complex, the refinement, and the balancing condition should appear in the main text. Perhaps use the median example in Section 2. For example, to understand the balancing condition, I had to draw a 3D cube, a trivial case of a 3-dimensional polyhedral complex. According to the definition, for any 2-dimensional face \sigma, we have a weight w(\sigma). Then, for any 1-dimensional face \tau and any 2-dimensional face containing \tau, we have the vector e_{\sigma/\tau}. The balancing condition says that if we sum up all these vectors scaled by their weight, we should get the zero vector.

Page 4: Why is the balancing condition required only for n-2 dimensional faces? Why do we not care about smaller faces?

Page 4: It’s probably easy, but I couldn’t see it. Can you provide an example of a convex CPWL function with two different compatible partitions?

Page 4, line 188: Give an example of the number of pieces. E.g., the number of pieces for the median function is 6, correct?

Page 4, line 188: Same for affine components. For the median, it is 3, correct? Also, in terms of notation, shouldn’t that better be \(\text{aff}(\mathcal{P})\) or something like that?

Page 4: What is the relationship between k, q and n? Knowing this is also important for the representation results in Section 6.

Page 5, line 241: Recall that w are weights for codimension 1 faces.

Page 5, line 247: You write that Figure 1 illustrates the different parameterizations of the median function according to Lemma 3.2. I was trying to understand what that means. I guess the numbers on the figure are the weights on each edge, and the only “n-2” face for which we need to check the balancedness conditions is the origin, and this is indeed balanced. But how does that illustrate the isomorphism from Lemma 3.2?

Page 5, line 250: Have you defined “regular complex”?

Page 5, definition 3.4: Could it be that although f is compatible with P, it has a DC decomposition where f and g are not compatible with P? That is, are the decompositions studied here a restricted class?

Page 6, definition 3.12: I forgot what was meant by “pieces” here? It is the minimum number of facets in a compatible decomposition. Apparently, this is already used in the literature, but maybe a better term could be used?

Page 6, line 302: Typo.

page 6, Theorem 3.13. I may have missed this but have you argued that the decomp polyhedron always has an extreme point?

Page 7, proposition 4.1: The word “function” appears twice.

Page 7, line 371: What is proposition B.3?

Page 9, line 468: Typo “certing.”

Page 9, line 450: “this simple fact…” it is not clear what that means, please explain. Are Theorems 6.1 and 6.2 constructive? Would we be required to know the parameters q and k and their specific representations as CWPL functions?

Page 9, Theorem 6.3: Writing q = k in the theorem statement makes it look like an assumption (which is not the case).

Page 10, Corollary 6.4. This result is only applicable to CPWL functions that are compatible with a regular polyhedral complex. Maybe I missed it, but can you discuss whether this assumption is natural and/or easy to satisfy.

---

> ### Author Response · Authors · 2024-11-20
> **1/3**
>
> We thank the reviewer for the careful examination of our manuscript and their thoughtful comments. We hope to have addressed all items raised in the revision and the responses below. If there are any further questions, please feel free to post them and we are happy to clarify them.
>
> *The assumption that the DC decomposition should be with respect to a fixed polyhedral complex seems restrictive and perhaps unmotivated*
>
> We agree with the reviewer that this assumption imposes a restriction, as discussed in our limitations section. However, we think the assumption is well-motiviated from multiple perspectives. Firstly, finding a minimal decomposition without such an assumption seems to be a very challenging problem, so the assumption seems to be needed to develop useful theory. Secondly, the assumption is motivated through our application to the theory of submodular functions, where it appears naturally through the concept of the Lovasz extension as discussed in Section 5. And thirdly, the assumption is also motivated through previous theoretical works in neural network theory, see e.g. Hertrich et al. (NeurIPS 2021), where a similar assumption was used to study lower bounds on neural network depth.
>
> *I feel that the paper is hard to read for non-experts in the area (e.g. many technical definitions with not many accompanying figures to help the reader-there are few, but mostly in the Appendix).*
>
> We thank the reviewer for raising this point including all the specific feedback and comments below. As a theoretical paper comprising different disciplines and being restricted to 10 pages, a certain level of complexity is unavoidable in writing. We gave our best to implement the concrete suggestions of the reviewer to make the paper more readable, where possible within the page limit.
>
> *The main result in Section 6 (Corollary 6.4) for representing CPWL functions as NNs, is only applicable to CPWL functions that are compatible with a regular polyhedral complex. It is not clear whether this assumption is (1) natural and (2) easy to satisfy. The existing decomposition results are applicable to any CPWL function.*
>
> We would like to emphasize that every CPWL function is compatible with *some* regular polyhedral complex. This is, for example, used in the cited construction by Hertrich et al. (2021) for the non-convex case. The only question is how many pieces this complex has. The fewer pieces in the regular complex associated with your CPWL function, the stronger Corollary 6.4 is. In so far, the assumption is "easy to satisfy" somehow, the interesting (and largely open) question is, with what complexity. Regarding how natural the assumption is, we would like to refer to the discussion above as well as to the examples towards the end of Section 3.
>
> *Page 1, Introduction: You write, “CWPL functions play a crucial role in ML.” If I understand correctly, this is the case because they are used as a test case for the universality theorems for NNs, where they can be concretely instantiated in terms of width, depth, etc. Is that correct, or are there more important applications? You do mention submodularity etc in the introduction, maybe elaborate on that somewhat?*
>
> The most important role of CPWL functions in ML is, in our opinion, that a function can be represented by a ReLU neural network if and only if it is CPWL, by Arora et al. (2018), as we cite immediately in the next sentence. Thus, every study about the expressive power of such networks is essentially a study about certain CPWL functions, and we cite many examples in our paper. Submodularity is, of course, another strong motivation, and is largely explained through our Section 5.
>
> *page 1: Are there problems where fixing the supporting polyhedral partition in DC decompositions is a natural constraint?*
>
> Yes, as mentioned above, this comes in naturally in the study of submodular functions, whose Lovasz extensions are always compatible with the complex induced by the braid arrangement, see Section 5. It is also used in theoretical work about neural network expressivity, see Hertrich et al. (2021).

---

> ### Author Response · Authors · 2024-11-20
> **2/3**
>
> *Page 2, Section 1.1: Please point to a specific Theorem as your “main technical result.” I guess that is Theorem 3.13?*
>
> We would not want to call one of our theorem *the* main result, as the interplay between the different perspectives and subjects is a crucial theme in our paper. However, to improve readibility and see interconnections, we added references to some theorems in Section 1.1.
>
> *Page 2, line 73: “cannot be easily simplified.” Explain why this is important. Also, when you say a “minimal” solution, what are you referring to? Page 2, line 75: “simply enumerating” seems to indicate there are not that many of them. Please elaborate on this.*
>
> If you have a decomposition $f=g-h$ and there is a convex function that you can subtract from both $g$ and $h$ without making them nonconvex, then this implies that your decomposition is trivially suboptimal. This is what is meant by "easily simplifying". Explaining this would make this short introductory paragraph too long, but it is explained in Section 3. "Simply enumerating" does not mean there are not many of them, it only means that the algorithm is "simple" (basically brute force). We changed the sentence and highlighted in the writing that it could be many vertices.
>
> *page 3. Please give a bivariate example of a CPWL function with two different compatible polyhedral complexes with a different number of maximal facets.*
>
> For this example, we would like to refer to Example 1 and Figure 1d  in https://arxiv.org/pdf/2205.05647v4 .
> This example considers the function
> $$
> \psi_1(x) = \begin{cases}
>     x, & 0 \leq x \leq y \newline
>     y, & 0 \leq y \leq x \newline
>     0 & \text{otherwise}
> \end{cases}.
> $$
> Figure 1d displays two incomparable compatible polyhedral complexes:
> The union of V(g1) and V(h1) yields a complex which defined by the rays from the origin through the points (1,0), (0,-1), (-1,0), (0,1), (1,1),
> whereas the union of V(g2) and V(h2) yield the complex defined by the rays through the origin and the points (1,0), (-1,-1), (0,1), (1,1).
> The first complex has 5 2-dimensional faces, the second complex has 4 2-dimensional faces, and both are compatible with the function $\psi_1$.
>
>
> *Page 4: Please provide a simple example (with a figure) of a polyhedral complex illustrating (some of) the definitions in Section 2. In particular, at least one example of a polyhedral complex, the refinement, and the balancing condition should appear in the main text. Perhaps use the median example in Section 2. For example, to understand the balancing condition, I had to draw a 3D cube, a trivial case of a 3-dimensional polyhedral complex. According to the definition, for any 2-dimensional face \sigma, we have a weight w(\sigma). Then, for any 1-dimensional face \tau and any 2-dimensional face containing \tau, we have the vector e_{\sigma/\tau}. The balancing condition says that if we sum up all these vectors scaled by their weight, we should get the zero vector.*
>
> We added the balancing condition to the caption of Figure 1a, and added pointers to Figures 1a and 1b at appropriate points in Section 2.
> A refinement of this polyhedral complex is given e.g. by the function which computes the second largest value of $\{0,x_1,x_2,x_1+x_2\}$, whose supporting polyhedral complex has an additional hyperplane with normal $(1,1)$, subdividing the second and fourth quadrant, thus yielding a polyhedral complex with 8 maximal, 2-dimensional polyhedral cones.
>
> *Page 4: Why is the balancing condition required only for n-2 dimensional faces? Why do we not care about smaller faces?*
>
> This is a classical result in tropical geometry (see the structure theorem in Maglagan & Sturmfels (2015)). We transfer this to our setting in Lemma 3.2. We added a pointer after the definition of the balancing condition.
>
> *Page 4: It’s probably easy, but I couldn’t see it. Can you provide an example of a convex CPWL function with two different compatible partitions?*
>
> As mentioned at the end of Section 2, every convex CPWL function f has a unique coarsest compatible complex $\mathcal P_f$, so any other complex compatible with $f$ is a coarsening of $\mathcal P_f$. A simple example is given by the function $f(x,y) = max(x,y)$, where $\mathcal P_f$ consists of a single hyperplane defined by the equation $x=y$. A coarsening $\mathcal P$ (which is, in particular, also compatible with $f$) is given e.g. by adding the additional hyperplane $x = -y$. This yields a polyhedral complex which satisfies $\mathcal P = \mathcal P_g$ for $g(x,y) = max(2x, x-y, x+y, 0)$.
>
> *Page 4, line 188: Give an example of the number of pieces. E.g., the number of pieces for the median function is 6, correct?*
>
> Correct, we added this to the caption of Figure 1.

---

> ### Author Response · Authors · 2024-11-20
> **3/3**
>
> *Page 4, line 188: Same for affine components. For the median, it is 3, correct? Also, in terms of notation, shouldn’t that better be (\text{aff}(\mathcal{P})) or something like that?*
>
> We also added k=3 to the caption of Figure 1. We restrain to use the notation $\text{aff}(\cdot)$ due to its reminisence to $\text{Aff}(\cdot)$, which denotes the smallest affine subspace containing $\cdot$.
>
> *Page 4: What is the relationship between k, q and n? Knowing this is also important for the representation results in Section 6.*
>
> The relationship is $n < k \leq q \leq k!$, as discussed on p.4, l.194-l.195.
>
> *Page 5, line 241: Recall that w are weights for codimension 1 faces.*
>
> In Lemma 3.2, w is defined as a map from $\mathcal{P}^{n-1}$ to $\mathbb{R}$ and therefore we believe that it is apparent in the statement that w are weights for codimension 1 faces.
>
> *Page 5, line 247: You write that Figure 1 illustrates the different parameterizations of the median function according to Lemma 3.2. I was trying to understand what that means. I guess the numbers on the figure are the weights on each edge, and the only “n-2” face for which we need to check the balancedness conditions is the origin, and this is indeed balanced. But how does that illustrate the isomorphism from Lemma 3.2?*
>
> We added some explanation to the figure.
>
> *Page 5, line 250: Have you defined “regular complex”?*
>
> The definition of a regular polyhedral complex is given on p.4, l.202.
>
> *Page 5, definition 3.4: Could it be that although f is compatible with P, it has a DC decomposition where f and g are not compatible with P? That is, are the decompositions studied here a restricted class?*
>
> Yes, it depends on the choice of $\mathcal{P}$. See also Proposition B.3, where we provide an example of a decomposition that is not compatible with the hyperplane extension of a function f.
>
> *Page 6, definition 3.12: I forgot what was meant by “pieces” here? It is the minimum number of facets in a compatible decomposition. Apparently, this is already used in the literature, but maybe a better term could be used?*
>
> By pieces we mean the number of maximal polyhedron in the unique coarsest polyhedral complex compatible with the convex function g, as defined in Section 2.
>
> *Page 6, line 302: Typo.*
>
> done.
>
> *page 6, Theorem 3.13. I may have missed this but have you argued that the decomp polyhedron always has an extreme point?*
>
> In general, a polyhedron has a vertex if and only if it does not contain an affine space. For decomposition polyhedra, this follows immediately from Theorem 3.5.
>
> *Page 7, proposition 4.1: The word “function” appears twice.*
>
> done.
>
> *Page 7, line 371: What is proposition B.3?*
>
> Proposition B.3 can be found in Appendix B, and states that there exists a CPWL function $f$, and convex CPWL functions $g,h$ with $f=g-h$ such that every decomposition $(g',h') \in \mathcal{D}_{\mathcal{H}_f}(f)$ is dominated by $(g,h)$.
>
> *Page 9, line 468: Typo “certing.”*
>
> done.
>
> *Page 9, line 450: “this simple fact…” it is not clear what that means, please explain. Are Theorems 6.1 and 6.2 constructive? Would we be required to know the parameters q and k and their specific representations as CWPL functions?*
>
> It refers to the fact from the previous sentence, which is the result that the maximum of $n$ numbers can be computed with depth $\lceil \log_2n \rceil+1$ and overall size $\mathcal{O}(n)$.
> The Theorems are constructive. We would need to know the representation of Wang & Sun (for Theorem 6.1) or the lattice representation (for Theorem 6.2) since the Theorems deduce the representation as neural networks from these known representations of CPWL functions.
>
> *Page 9, Theorem 6.3: Writing q = k in the theorem statement makes it look like an assumption (which is not the case).*
>
> We moved the 'q=k' to the conclusion of the statement.
>
> *Page 10, Corollary 6.4. This result is only applicable to CPWL functions that are compatible with a regular polyhedral complex. Maybe I missed it, but can you discuss whether this assumption is natural and/or easy to satisfy.*
>
> Every CPWL function f is compatible with a regular polyhedral complex, e.g. given by the hyperplane extension (Construction B.1). However, it is unknown if one can find a regular complex compatible with f such that the number of full-dimensional polyhedra in this complex is not much larger than the number of pieces q of the function f.

---

### Meta-Review · Area_Chair_SqrE · 2024-12-19

**Metareview:**

The paper studies the problem of decomposing a continuous piecewise-linear (CPWL) function into a difference of two *convex* CPWL functions. The motivation is that CPWL functions can represent all possible ReLU neural networks (and vice-versa), while an explicit representation of a nonconvex CPWL function as a "difference of convex" (DC) CPWL function allows for usage of extensive theoretical and algorithmic framework in the domain of optimizing DC functions.

The paper first identifies an assumption under which the studied problem is tractable. It then proves a series of structural results, of which the highlights are that: (1) there is a polyhedron with the property that all irreducible solutions are faces while minimal solutions are vertices (so a minimal DC CPWL decomposition can be found by enumerating the vertices) and (2) under an additional assumption provided in the paper, the vertex corresponding to the minimal decomposition is unique. The paper further discusses connections to and implication on the theory of submodular functions.

The paper is technically solid and of high interest to the ML community.

**Additional Comments On Reviewer Discussion:**

Most questions raised in the reviews were clarification questions; there were no major concerns identified in the reviews. The authors responded appropriately to all the questions and revised the paper.

---

### Decision · Program_Chairs · 2025-01-22

Accept (Spotlight)